# MolMiner: Transformer architecture for fragment-based autoregressive generation of molecular stories

## Abstract

Deep generative models for molecular discovery have become a very popular choice in new high-throughput screening paradigms. These models have been developed inheriting from the advances in natural language processing and computer vision, achieving ever greater results. However, generative molecular modelling has unique challenges that are often overlooked. Chemical validity, interpretability of the generation process and flexibility to variable molecular sizes are among some of the remaining challenges for generative models in computational materials design. In this work, we propose an autoregressive approach that decomposes molecular generation into a sequence of discrete and interpretable steps using molecular fragments as units, a 'molecular story'. Enforcing chemical rules in the stories guarantees the chemical validity of the generated molecules, the discrete sequential steps of a molecular story makes the process transparent improving interpretability, and the autoregressive nature of the approach allows the size of the molecule to be a decision of the model. We demonstrate the validity of the approach in a multi-target inverse design of electroactive organic compounds, focusing on the target properties of solubility, redox potential, and synthetic accessibility. Our results show that the model can effectively bias the generation distribution according to the prompted multi-target objective.

## 1 Introduction

Deep generative models (DGMs) have become a popular choice in new high throughput screening (HTS) paradigms (Westermayr et al., 2023; Ortega Ochoa et al., 2023). Within new HTS, generative models are used to create an initial pool of candidates subject to some target properties (Sanchez-Lengeling & Aspuru-Guzik, 2018). This initial pool is then filtered in sequential steps of increasing computational expense, from machine-learning surrogate models (Schütt et al., 2017; Tsubaki & Mizoguchi, 2020a) to classical computing methods, such as density functional theory (DFT; Kohn & Sham, 1965). These generative models vary in the molecular representation used, e.g., SMILES strings (Gómez-Bombarelli et al., 2018; Lim et al., 2018; Popova et al., 2018; Podda et al., 2020), graph-based (Jin et al., 2019; 2020) or point sets (Hoogeboom et al., 2022b; Gebauer et al., 2022; Guan et al., 2023; Schneuing et al., 2023; Qiang et al., 2023), and modelling approach, e.g., reinforcement learning-based (Popova et al., 2018; Simm et al., 2020a;b), variational autoencoders (VAEs; Gómez-Bombarelli et al., 2018; Lim et al., 2018), generative adversarial network (GANs; Cao & Kipf, 2022), diffusion models (Hoogeboom et al., 2022b; Guan et al., 2023; Schneuing et al., 2023; Qiang et al., 2023; Wu et al., 2022; Huang et al., 2022; Xu et al., 2023), normalizing flows (Satorras et al., 2022) or flow matching (Dunn & Koes, 2024). Despite the diversity of models, there are challenges unique to computational materials modelling that have been often overlooked:

**(A) Multi-step vs one-shot generation:** Generating a molecule in a single step risks making the generation process opaque. Generating a molecule in sequential steps allows spending more computation per step and makes the generation more transparent. Multi-step generative models for materials include Molgym (Simm et al., 2020a), diffusion models (Hoogeboom et al., 2022b; Qiang et al., 2023), flow matching (Dunn & Koes, 2024), VAE-based models with an autoregressive decoder (Jin et al., 2019; 2020), and other autorregressive

models (Gebauer et al., 2022; You et al., 2018; Liao et al., 2020; Xie et al., 2021; Maziarz et al., 2024).

**(B) The size of the molecule is fixed during the generation process:** A number of models that do satisfy the desirable feature **(A)** fixed the size of the molecule during the generation process. We believe the size of the molecule should be a choice of the model during the generation process. Some models that do take this into account include the graph-based, VAE variant models with autoregressive decoders (Jin et al., 2019; 2020), G-Schnet (Gebauer et al., 2022) which operates on point sets, MARS (Xie et al., 2021), MoLeR (Maziarz et al., 2024).

**(C) Low chemical validity of generated molecules:** Most often, chemical validity checks are enforced only at the end of the generation process, resulting in a lower percentage of chemical valid molecules. Incorporating chemical checks during the multi-step generation process allows to identify chemical violations before the molecule is completed, thus avoiding continuing the generation of chemically invalid molecules. Models that do satisfy this desirable feature are the graph-based AE variant models with autoregressive decoders JTNN, Hier-VAE (Jin et al., 2019; 2020), MARS (Xie et al., 2021), MoLeR (Maziarz et al., 2024).

**(D) Coarse-graining molecules:** Molecules naturally exhibit hierarchical structure, they often can be decomposed into molecular fragments that can be treated as units in themselves. Exploiting the hierarchical nature of molecules helps scale deep learning models to larger molecules (Wang & Gómez-Bombarelli, 2019), something that has remained a notable challenge. Models like JTNN and HierVAE use molecular fragments as units, and this approach has also been adapted for diffusion models (Qiang et al., 2023).

**(E) Incorporating 3D information:** A model that has no access to 3D geometry information of the molecule will struggle to conditionally generate other structures when the target property depends on the 3D geometry. Models like JTNN, HierVAE, MARS or MoLeR do not include this information in the molecule representation. This has remained a notable challenge for autoregressive models, as discussed in (Voloboev, 2024).

Our proposed model is designed to satisfy the five desirable requirements in the most general formulation we could think: a purely autoregressive (no encoder), semi-order-agnostic, multi-step, multi-property generation using symmetry-aware molecular fragments and 3D geometry of a non-fixed size molecule with enforced chemistry sanitation during generation.

## 1.1 RELATED WORK & CONTRIBUTION

Our work builds on ideas from the aforementioned studies, but it is most closely related to JTNN (Jin et al., 2019) and HierVAE (Jin et al., 2020), particularly the latter. HierVAE is a hierarchical graph VAE-based model operating on molecular fragments using an autoregressive decoder to build molecules in sequential steps. Similarly, we enforce chemical validity during the generation process, and, like JTNN and HierVAE, we use a coarse-graining procedure to extract molecular fragments. In summary, our contributions are:

- We propose a semi-order-agnostic autoregressive model that grows molecules in discrete steps.

- We formulate the one-step prediction step as a single classification task, as opposed to the nested classification approach of Jin et al. (2020).

- We formulate an approach to uniquely identify attachments of a fragment taking into account its symmetries.

- We show how to incorporate spatial information in the autoregressive process to make the prediction geometry-aware.

- We demonstrate the model can be used for multi-target inverse design by examining how well calibrated are the predicted properties of compounds generated subject to design criteria spanning a whole dataset.

---

**Algorithm 1** Extract fragments from Molecular Graph $\mathcal{M}$

---

1: **function** GENERATEFRAGMENTS($\mathcal{M}$)
2:     $fragments \leftarrow [\text{TUPLE}(x) \textbf{ for } x \textbf{ in } \text{GETSSSR}(\mathcal{M})]$
3:     **for each** $bond$ **in** $\mathcal{M}$.GETBONDS() **do**
4:         $a_1, a_2 \leftarrow bond$.GETBEGINATOM()$, bond$.GETENDATOM()
5:         $bond\_in\_existing\_fragments \leftarrow$ **False**
6:         **for each** $fragment$ **in** $fragments$ **do**
7:             **if** ($a_1$ **in** $fragment$) **and** ($a_2$ **in** $fragment$) **then**
8:                 $bond\_in\_existing\_fragments \leftarrow$ **True**
9:                 **break**
10:        **if not** $bond\_in\_existing\_fragments$ **then**
11:           $fragments$.APPEND($(a_1, a_2)$)
        **return** $fragments$

---

## 2 METHODS

### 2.1 COARSE GRAINING MOLECULES

Molecules exhibit hierarchical structure, and we often find repeating fragments within a molecule that can be treated as units in their own right to obtain a coarse representation of the molecule. A coarse representation is particularly useful because it simplifies the problem by shifting the focus to global patterns rather than the finer details. However, there is not a unique hierarchy. There are a variety of ways of decomposing a molecule into constituent fragments. The choice of the decomposition procedure depends on the application and the level of resolution at which one wants to describe the molecule. Nevertheless, there are some desirable requirements for the fragments and the decomposition procedure to satisfy. **Uniqueness**: A molecule should always be decomposed into the same set of fragments that are themselves irreducible. **Disjoint fragments**: A molecule is decomposed into fragments that do not overlap. So that the whole molecule can be reconstructed by docking fragments with one another. **Interpretable fragments**: The fragment constituents of the molecule are commonly used chemical fragments, for a better synchrony between the chemist and the model. Following from these three self-imposed requirements, we propose the procedure described in Algorithm 1. The procedure finds the Smallest Set of Small Rings (SSSR) of a molecule and uses them to segment the molecule into its rings and all other individual bonds not belonging to a single ring.

### 2.2 STANDARDIZATION OF FRAGMENTS AND ATTACHMENTS

A fragment of a molecule is the irreducible unit extracted using Algorithm 1, and its attachment points are the fragment's atoms that the neighbour fragments dock to. Using the fragments and its attachment points as building blocks requires a unique way of representing them. A natural, unique and human-readable representation for the fragments are their Canonical SMILES strings (Weininger, 1988; Weininger et al., 1989). However, a Canonical SMILES string can not encode the attachment information. Given a molecule, all the atoms in said molecule have an index number associated. This atom index can be used for identifying the atoms that serve as attachments between fragments. In Figure 1, there are three fragments: a six-membered ring with atoms $(3, 4, 5, 6, 7, 8)$, a five-membered ring with $(1, 2, 3, 8, 9)$ and a bond-type fragment with atoms $(0, 1)$. The two ring fragments dock at atoms $(3, 8)$ and the five-membered ring docks with the bond-type fragment at $(1)$. This atom-indexing is referred to as 'global indexes'. When the fragment is extracted as a separate entity, the atom-index numbering is automatically reset. However, we can keep track of the map between the 'global' index and the new 'local' index as we extract the fragment, retaining the docking point information. Through this map, we know the 6-membered ring is docked with the 5-membered ring at local indexes $(4, 5)$. This fragment is then encoded as the Canonical SMILES string to obtain a unique representation of the fragment that is invariant to atom indexing.

We want to express the attachment points in the local index system of the Canonical fragment so they are uniquely identified. The challenge is that when loading the graph from the Canonical SMILES, unless the original fragment was itself in the canonical index form, we have once again a new 'local'

---

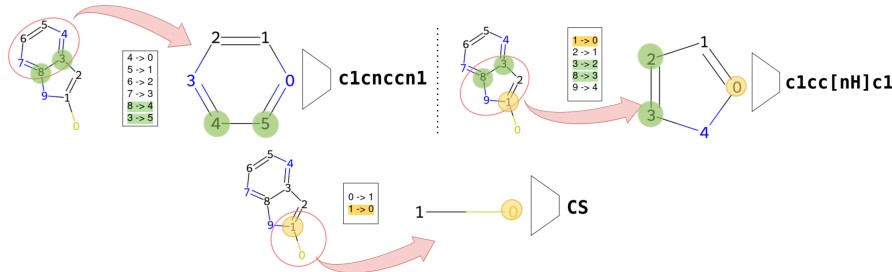

Figure 1: Extraction of fragments and attachments from a molecule. For each, fragment we use the map between the 'global' and 'local' indexes to track which of the fragment's atoms participated in attachments. The fragment extracted is then encoded into its Canonical SMILES.

index, and because the SMILES removes all atom-index information we lose the map between previous 'local' atom indexes and the 'local' indexes in the canonical fragment. This problem is illustrated in Figure 2 (a). This 'lost map' can, however, be recovered by exploiting how SMILES strings are generated and the specific choice of fragments decomposition described in Algorithm 1. The fragments resulting from Algorithm 1 are single-cyclic graphs (single rings, or bonds, which are '2-membered rings'). When a Canonical SMILES string is created for these fragments, the encoding algorithm traverses the graph alongside the single cycle, which results in any possible new indexing being a cyclic permutation of the original indexing. Computing the similarity matrix based on Tanimoto distance of the Morgan fingerprints (Rogers & Hahn, 2010) from each of the atoms for the canonical and non-canonical fragment, we can see all possible maps superposed. Using the prior knowledge that any remap will be a clock shift, we can extract all shifts (positive or negative) that will allow a map from the canonical to the non-canonical fragment. This is illustrated in Figure 2.

Due to the symmetries of the fragments, we find multiple possible maps. In the case of Figure 2, there are four such maps. Any of these maps can be used to find the attachment points in the canonical fragment (highlighted in green), so we choose to use one, the first one, as the map between the non-canonical fragment and the canonical fragment. Then we can say that there is a 6-membered ring fragment that in the global system has atoms $(3, 4, 5, 6, 7, 8)$ and an attachment at $(3, 8)$ or in its canonical local system it has atoms $(0, 1, 2, 3, 4, 5)$ and has attachments with another fragment at $(4, 3)$. This procedure works to decompose the molecules into canonical fragments and attachments. The fragments are represented by the Canonical SMILES and the attachments are represented as the atom indexes of an attachment in the canonical form, making them unique. However, if we take into account the symmetries, this definition of standard attachments is problematic. Following the previous example, attachments at the atoms $(4, 3)$ or $(1, 0), (0, 1), (3, 4)$ are indistinguishable.

To fix this, we create the standardization map. The standardization map lets us check if any two canonical attachments are the same. To construct this standardization map we take one of the reconstructed maps (e.g., the first one) as reference, invert it, and convolute it with all other maps. For each item, its standard map is the minimum of the values of the convoluted maps. For example, atom 0 from the reference canonical form can be mapped on to $0, 3, 4, 1$, then since 0 is the minimum in the standard map 0 maps to 0. If we take now atom index 2 (which in the reference canonical form corresponds to a nitrogen), it can be mapped to $2, 5, 2, 5$, of which the minimum is 2, so in the standard map 2 maps to 2. In the appendix, Figure 7 fully illustrates this example. Using the standardization map, we can confirm that $(3, 4), (4, 3), (1, 0)$ and $(0, 1)$ are the same type of attachment because all of them map to the tuple $(0, 0)$ using the standardization map. Then, we will always take one as reference, the first one ever encountered, e.g., $(0, 1)$, and whenever the attachment is seen as a canonical attachment $(3, 4)$ we will say its a $(0, 1)$ standard attachment.

### 2.3 UNROLLING MOLECULAR STORIES

We model molecular generation as a semi-order-agnostic autoregressive process. This requires a procedure for unrolling a molecule into a sequence of attachment steps, which we call a molecular story. A molecular story $S$ starts with a single fragment to which other fragments are attached until the molecule is completed. A story can start from any fragment and can be grown in any particular

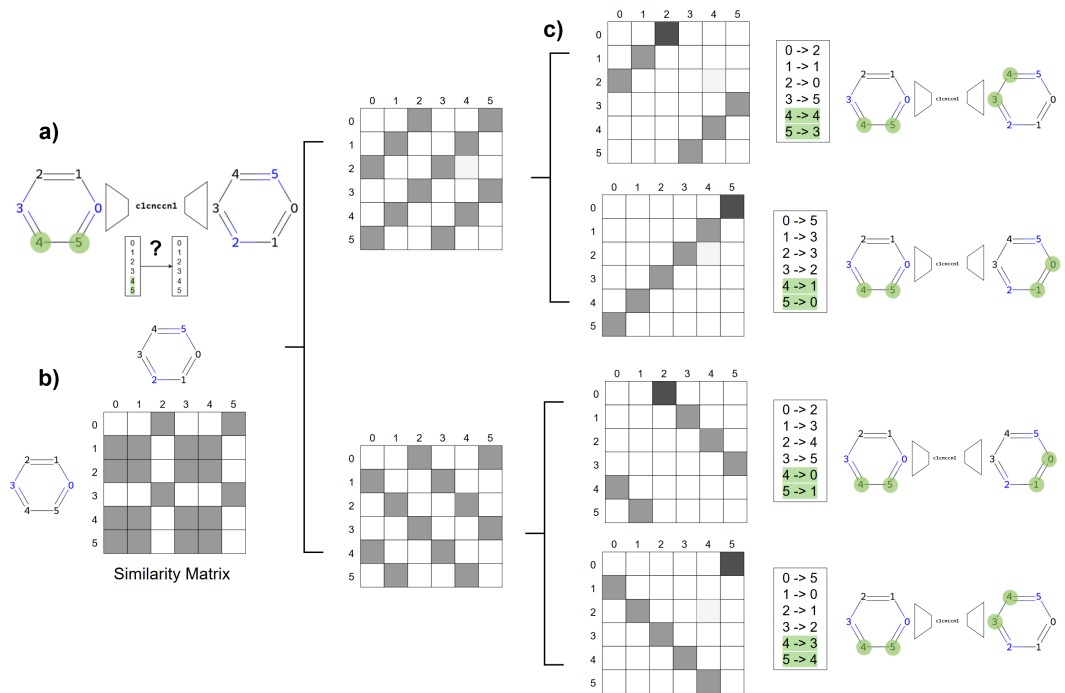

Figure 2: Extraction of all the possible maps between 'local' indexes. After a fragment is encoded into its SMILES if we then re-create the fragment from said SMILES the 'local' index changes and the map between the old and new 'local' indexes is unknown **(a)**. This lost map can be recovered by computing the Tanimoto similarity of Morgan fingerprints of every pair of atoms in the fragment, resulting in a similarity matrix **(b)**. Using this similarity matrix, and the prior knowledge that any possible map will be a cyclic permutation of the indexes, we can extract all possible cyclic permutations allowed by the similarity matrix, which leads into all the possible maps from the old to the new 'local' index **(c)**.

order, only bound by having to dock new fragments to existing ones. In terms of graph theory, this is a graph traversing procedure where the exploration node frontier is sampled at random (as opposed to the First In First Out approach of BFS or the Last In First Out of DFS).

To create a story of a given molecule, we begin by randomly selecting a fragment of the molecule and then initializing the exploration queue by adding all chemically possible attachment points of the fragment in the form of tuples (fragment, attachment). At every step, a tuple (fragment, attachment) is randomly sampled from the exploration queue. This fragment is referred to as *focal fragment* $(f_{\text{focal}}, a_{\text{focal}})$ and becomes the point at which, through the attachment, the molecule will grow. Given this tuple, if the original molecule had a fragment $(f_{\text{next}}, a_{\text{next}})$ docked at this location that has not been added already, the next fragment is created and docked to the focal fragment, updating the exploration queue with the new frontier. That is, $(f_{\text{focal}}, a_{\text{focal}})$ is docked with $(f_{\text{next}}, a_{\text{next}})$. Otherwise, the focal fragment is *cauterized* at the attachment by not docking anything. Finally, the focal fragment is removed from the exploration queue. The process stops when the exploration queue is empty, meaning that all fragments have been added at the correct attachment points and all other attachment locations have been cauterized. The procedure is described in Algorithm 2. At every step in the story of a molecule, the atomic coordinates are computed using a classical Force-Field (Rappe et al., 1992; Halgren, 1996). The Force-field is used to do a conformer search and return the lowest energy conformer.

## 2.4 MODEL AND ARCHITECTURE

Inspired by order-agnostic autoregressive models (Uria et al., 2014; Hoogeboom et al., 2022a), we propose a *semi-order-agnostic* autoregressive model, in which we allow factorization only over valid

---

**Algorithm 2** Randomly decoding Molecular stories

---

1: **Input:** Molecular graph $\mathcal{M}$
2: **Output:** List of all steps $\mathcal{S}(\mathcal{M})$ in story of $\mathcal{M}$
3: $f \sim \mathcal{U}(\text{GETFRAGMENTS}(\mathcal{M}))$          ▷ Uniformly sample a fragment
4: $S \leftarrow [f]$          ▷ Initialize the story
5: $Q \leftarrow \{(f,a) \mid a \in \text{ATTACHMENTS}(f)\}$          ▷ Initialize the queue
6: **while** $Q$ is not empty **do**
7:      $(f,a) \sim \mathcal{U}(Q)$          ▷ Uniformly sample from queue
8:      $f_{\text{next}} \leftarrow \text{GETNEXT}(S, \mathcal{M}, (f,a))$          ▷ Get fragment at dock location
9:      **if** $f_{\text{next}} \notin S$ and $f_{\text{next}} \neq \emptyset$ **then**
10:          $S.\text{APPEND}(f_{\text{next}})$          ▷ Append to story
11:          $Q \leftarrow \{(f_{\text{next}}, a) \mid a \in \text{ATTACHMENTS}(f_{\text{next}})\}$          ▷ Update queue
         ▷ I.e., it is cauterize when $f_{\text{next}} = \emptyset$ or $f_{\text{next}} \in \mathcal{S}$
12:      $Q \leftarrow Q - \{(f,a)\}$          ▷ Remove explored sample
     **return** S

---

molecular stories. Specifically, we assume that the probability of a molecular graph $\mathcal{M}$ factorizes as

$$p(\mathcal{M}) = \mathbb{E}_{S \sim \mathcal{U}(\mathcal{S}(\mathcal{M}))} \prod_{i=1}^{\mathcal{O}_S} p_\theta(x_i^{(S)} | \mathbf{x}_{<i}^S), \tag{1}$$

where $S = [x_0^S, \ldots, x_{\mathcal{O}_S}^S]$ is a molecular story, $x = (f,a)$ is a tuple of fragment and attachments points, $\theta$ the model parameters, and $\mathbf{x}_{<i}^S$ is shorthand for $x_0^S, x_1^S, \ldots, x_{i-1}^S$. The conditional density in Equation (1) is based on decoder-only transformer (Vaswani et al., 2023) architectures, which are commonly used in language models. A schema of the model architecture is illustrated in Figure 3, and in Figure 4, the integration of the model in the story generation is shown.

In the architecture, molecular fragments are represented by learnable embeddings, the positional embeddings are removed, and the attention mechanism is modified to incorporate the spatial structure. The model first takes as input the collection of fragment embeddings, their local docking environment features referred to as docks saturation, and the target conditions. A fully connected layer is used on every fragment to embed its dock saturations and conditions into the embedding. The dock saturations are tuples of 3 elements, representing the percentage of 'docks in use', 'free docks', 'cauterized docks' of a fragment. These saturation features are scaled to the $[-1, 1]$ range before being fed to the network. The resulting tensor forms the input to a transformer block. Inside the attention heads, the self-attention mechanism is modified to bias the attention weights by introducing a discount factor product of a learnable scalar value $a$ and the pairwise euclidean distance between all fragments. The learnable scalar parameter $a$ acts as a weight on the effect of the geometry.

As with other transformer models, the output of a transformer layer is used as input to the next transformer layer in a process repeated for $N$ layers. In this case, only the last transformer layer is modified. There the query is the final embedding of the focal fragment, and the pairwise distance matrix is computed between all fragment positions and the position of the attachment. The result of this final layer is a tensor treated as a hidden representation of the next (fragment, attachment) to be added, which can then be projected into the vocabulary. However, to aid the model in deciding what to add next, the final projection layer also takes as input the docks saturation of the focal fragment, a learnable embedding of the type of attachment and the target conditions. The output is a probability distribution over the vocabulary formed by the combination of fragments and attachment points $\{(f,a) \mid f \in \mathcal{V}_f \text{ and } a \in \mathcal{V}_a(f)\}$.

## 2.5 TRAINING

We wish to learn the model by maximum likelihood estimation. However, we use a lower bound on the likelihood that is more amenable to stochastic optimisation, similar to order-agnostic autoregressive models (Uria et al., 2014; Hoogeboom et al., 2022a). For a single molecule $\mathcal{M}$, we can write the log-likelihood function as

$$\mathcal{L}(\theta|\mathcal{M}) = \log \mathbb{E}_{S \sim \mathcal{U}(\mathcal{S}(\mathcal{M}))} \prod_{i=1}^{\mathcal{O}_S} p_\theta(x_i^{(S)} | \mathbf{x}_{<i}^S) \geq \mathbb{E}_{S \sim \mathcal{U}(\mathcal{S}(\mathcal{M}))} \sum_{i=1}^{\mathcal{O}_S} \log p_\theta(x_i^{(S)} | \mathbf{x}_{<i}^S), \tag{2}$$

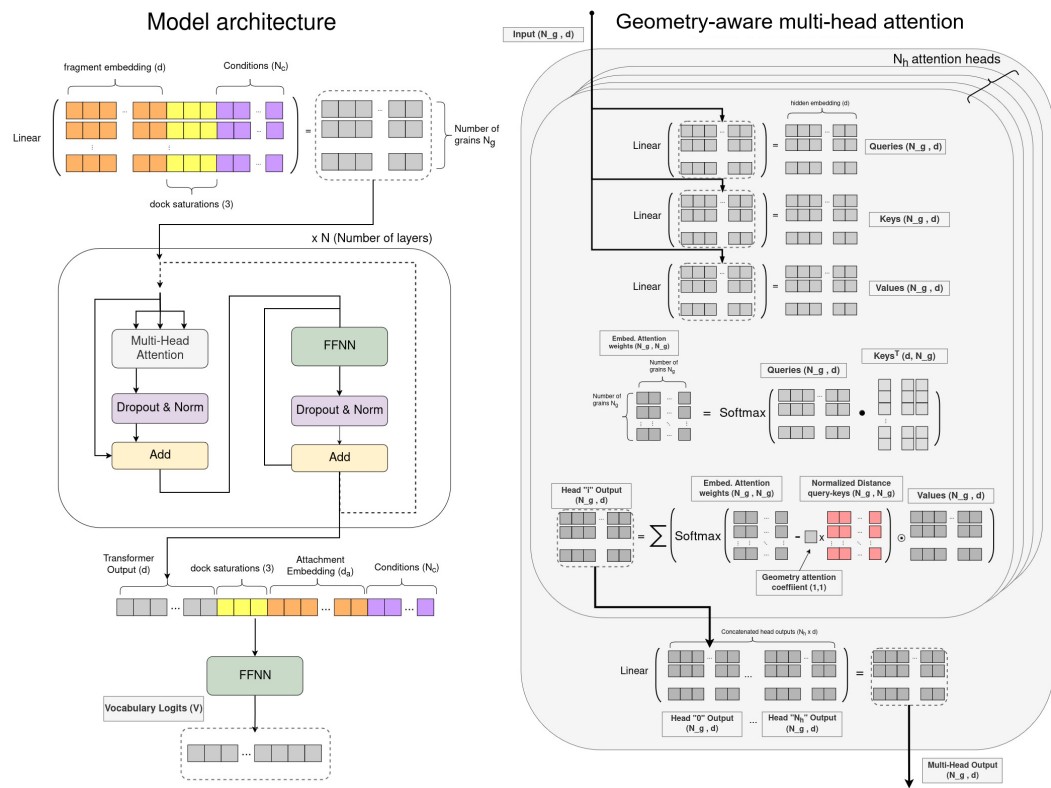

Figure 3: Molminer model architecture. On the left, the overall architecture. On the right, the geometry-aware multi-head attention. The spatial information is fed in as a tensor of pairwise distances (in red) weighted by a learnable scalar value.

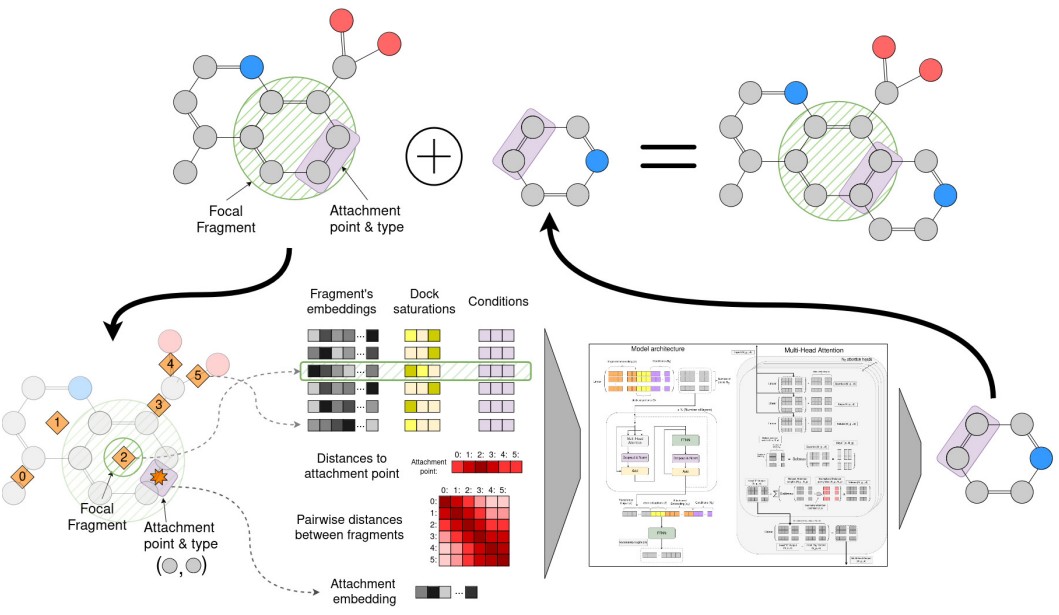

Figure 4: Integration of Molminer in the creation of a molecular story step. The model takes the molecule to be grown at a focal fragment through an attachment point and predicts which fragment at which attachment configuration should be added, resulting in a step of a 'molecular story'.

where the lower bound is obtained through Jensen's inequality (Jensen, 1905). In practice, this means that given a molecule $\mathcal{M}$, we sample a story $S \sim \mathcal{U}(\mathcal{S}(\mathcal{M}))$ of $\mathcal{O}_S$ steps and prompt the model at each story step what should be attached next. The objective is to maximize the probability of selecting the true next $(f_i, a_i)$ to be docked at $(f_{i-1}, a_{i-1})$. It must be noted that a molecule does not have a single story, so to allow the model to learn to grow molecules independent of any particular story, at every epoch for every molecule a new story is created. This also serves as data augmentation. This training procedure starts from an initial choice of a starting fragment. During training one of the target molecule building fragments is sampled randomly to start the molecular story. However, when generating new compounds the starting fragment is a choice on itself. To perform this task a separate simple model is created, whose architecture is a simple FFNN mapping some properties to the fragments vocabulary. The model's objective is to predict which of the available fragments will manifest in a molecule from its target conditions, a multi-class classification task. To train this model, the same train and validation split is used as with the autorregressive model, here, we aim to predict the set of fragments present in a molecule given its properties by minimizing the binary cross-entropy loss.

## 3 RESULTS

Molminer is trained on the RedDB dataset (Sorkun et al., 2022) of organic compounds for aqueous redox flow batteries. The dataset used includes $12\,185$ reactant molecules. Of those, $8\,529$ ($\sim 70\%$) are randomly selected to be used in training, and $3\,556$ ($\sim 30\%$) are reserved for testing. The remaining 100 molecules were used for validation. The targeted conditions are the reactant log-solubility, the redox potential, and the synthetic accessibility score (SAScore; Ertl & Schuffenhauer, 2009), the latter is not part of the dataset but is calculated for all the molecules. In addition to the dataset, AqSolPred (Sorkun et al., 2021) and QuantumDeepField (Tsubaki & Mizoguchi, 2020a;b) are used to act as surrogate models in predicting solubility and redox potential, respectively, of molecules outside the dataset. AqSolPred was already trained on this same dataset, whereas QuantumDeepField has been trained in-house on RedDB.

Molminer's hyperparameters were optimized using a reduced dataset containing 100 molecules from the training set and the 100 molecules from the validation set. A gridsearch was performed for fragment embedding size $(64, 128, 256)$, attachment embedding size $(16, 32, 64)$, number of attention heads per layer $(4, 8, 16)$, and number of transformer layers $(2, 3, 4)$ resulting in 81 models trained for 100 epochs or equivalently $10\,000$ stories. The rest of the parameters were left fixed at a learning rate of $10^{-4}$, the Adam parameters $(\beta_1, \beta_2) = (0.9, 0.9)$ and $\epsilon = 10^{-9}$, 512 hidden nodes in the fully connected layers, a dropout rate of 0.3, and an initialization of 1.0 for the geometry weight. Among the 81 resulting models, the best-performing model was found to have an accuracy of $80.93\%$ and $80.09\%$ on the training and validation sets. This model used 256 for the fragment embedding size, 64 for the attachment, 8 heads and 3 transformer layers. Using these hyperparameters, the model was trained on the entire training set. The model achieved $81.96\%$ and $82.94\%$ fragment-level reconstruction accuracy on the training and testing sets, respectively, after 80 epochs or equivalently $682\,320$ stories.

### 3.1 MOLECULAR GENERATION

To generate a compound, we give the model some desired properties, in this case, a desired redox potential, log-solubility and synthetic accessibility score and use the fragment initializer to return the probabilities of each of the fragments in the vocabulary manifesting in a molecule with such properties. Then one of the top-$k$ fragments with highest probability is uniformly sampled and used as the first building fragment from which the whole rest of the molecule is grown. The best results where achieved with at least $k = 3$; this experiment is discussed in the appendix. During the autoregressive generation, the fragments are sampled weighted by their probabilities, and chemistry rules are enforced at every step so that all attachments are valid.

To asses whether the conditional generation is calibrated, that is, the model creates compounds that obey our target properties, we create three separate experiments, one for each of the target properties. In each experiment, two of the target conditions are fixed at their mean, and we vary the remaining one for the range of values spanning from its minimum to the maximum of the dataset in 30 steps. For each of these 30 steps 30 molecules are generated with the same target condition in order to

have a estimate of the distribution of predicted properties for the same prompt. Within each unique prompt, we only take into account unique generations, removing all molecules that have already been created for that same prompt, and we remove any molecule that is consisting of a single fragment. Finally to have a notion of how far out of the dataset distribution a particular prompt is, a kernel density estimation (KDE) with Gaussian kernel and bandwidth $0.14$ is fitted to the distribution of properties in the dataset, so that any new sample prompt can be evaluated in terms of its score to how close or far it is of the dataset distribution. Then for each calibration plot we have: the scatter plot of prompted vs predicted property, the aggregated mean and standard deviation across the 30 generation of a particular prompt, and on a secondary plot the density of a the prompted sample evaluated by the KDE fitted on the dataset.

In order to benchmark our model, we perform this same procedure to a modified version of HierVAE (Jin et al., 2020). To see further details on the modification and other benchmarking results see the apendix. The results of the calibration experiments are shown in Figure 5 in red for HieVAE, blue for MolMiner. For each of these experiments, we compute the novelty ratio, the percentage of the molecules generated that are not in the dataset. For each unique prompt value, the predictions of the generated molecules are aggregated by computing the mean and standard deviation, shown in continuous and dashed blue/red lines respectively. The black dashed line represents the ideal correlation, where the predicted property is equal to the prompted property. For the three different experiments, the mean of the predictions for a given prompt follows the ideal correlation line (dashed black), but as we move to the tails of the distribution (as seen in the lower half of each sub-figure) the blue/red and black lines depart. These results shows that the models are well calibrated to perform multi-target inverse design for prompts within the dataset distribution, but as we query for target conditions where the dataset had scarce samples the models are not calibrated.

Table 1: Novelty percentage of the total generations for each of the three experiments, for each model. Note that Molminer generates a significantly higher percentage of novel molecules than HierVAE.

| Model | SAScore-exp | LogSolubility-exp | Redox Potential-exp |
|---|---|---|---|
| HierVAE-80epochs | 52.63 % | 55.56 % | 43.19 % |
| MolMiner-80epochs | 85.21 % | 79.19 % | 85.59 % |

From Figure 5 one can see that both HierVAE and MolMiner are well calibrated for conditional generation on each experiment. We want to emphasize that in each of these three experiments we are simultaneously optimizing the three properties, the only difference is that on each experiment we fix two of the properties values to be that of the mean of the dataset, while we vary the third one. This is done so we can visualize the calibration on each of these properties individually, as we would otherwise require a 6-dimensional plot. The three properties are always being simultaneously used. HierVAE is better calibrated than MolMiner for low Log-Solubilities and for high SAScores, while MolMiner is better calibrated for High redox potentials and lower SAScores, but both models perform otherwise comparably in terms of calibration to the target properties. However, when looking at the percentage of novel compounds generated we note a significant difference between the two models. Table 1 summarizes the novelty results on each of this experiments. For the three experiments, MolMiner is capable of generating significantly more novel molecules than HierVAE.

## 4 CONCLUSIONS

MolMiner is capable of producing significantly more novel molecules than its predecessor while subject simultaneously to three conditions. We attribute this performance enhancement to the removal of constraints in the autoregressive decoding by making the process semi-order-agnostic, to our handling of fragments symmetries, and the inclusion of the 3D geometry in the generative process.

The choice of the three simultaneously-imposed conditions used for conditional generation: Solubility, Redox Potential and Synthetic Accessibility, demonstrates the real world potential of our generative model in the search of novel electroactive organic compounds for aqueous redox flow batteries, a promising alternative for sustainable energy storage. Moreover we highlight that the

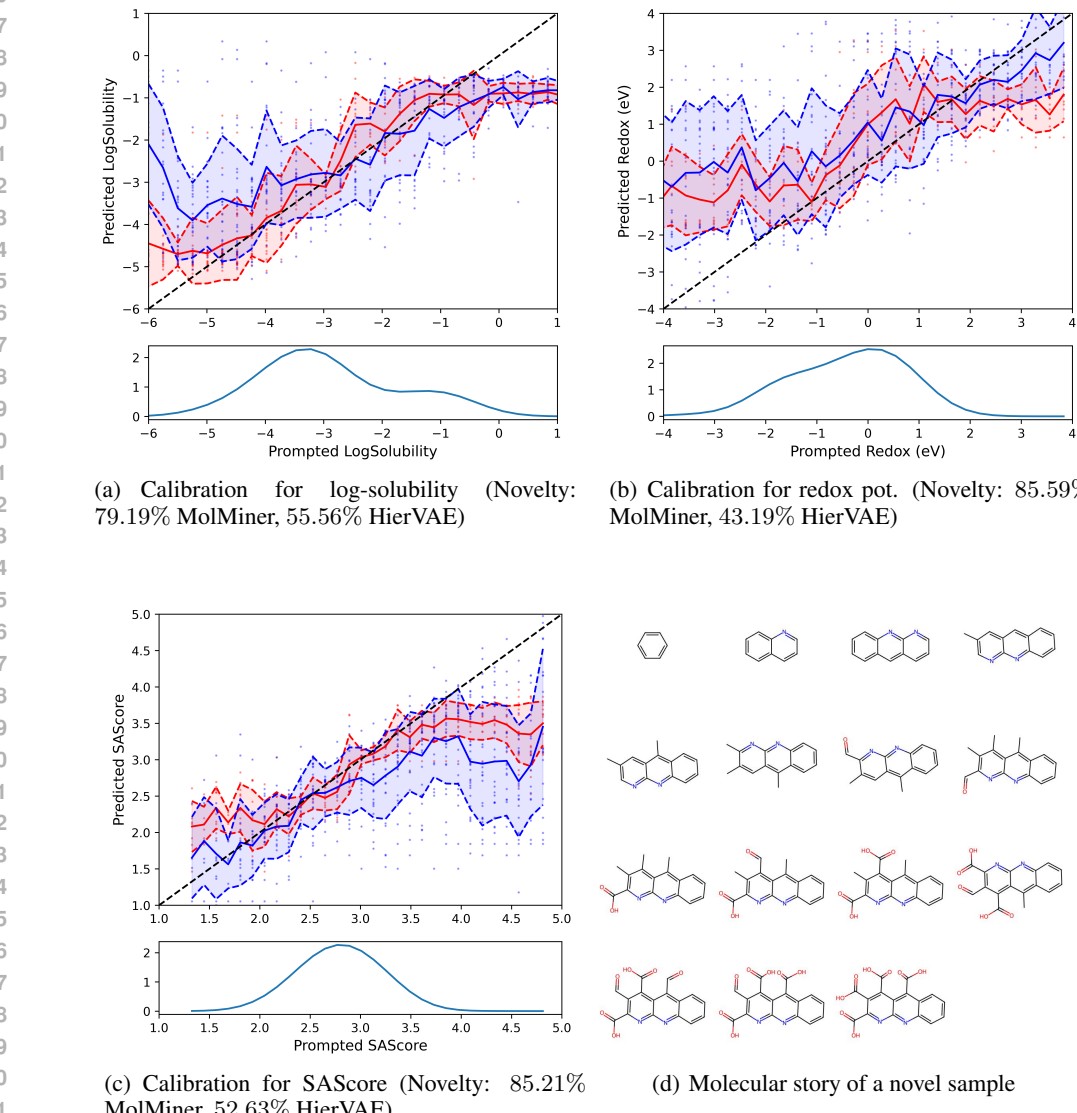

(a) Calibration for log-solubility (Novelty: 79.19% MolMiner, 55.56% HierVAE)

(b) Calibration for redox pot. (Novelty: 85.59% MolMiner, 43.19% HierVAE)

(c) Calibration for SAScore (Novelty: 85.21% MolMiner, 52.63% HierVAE)

(d) Molecular story of a novel sample

Figure 5: Calibration plots (red: HierVAE-80epochs, blue: MolMiner-80epochs) prompted vs predicted condition for the three properties trained on: (a) log-solubility, (b) redox potential, and (c) SAScore. The lower half of the figures represents the density of the KDE of each of the prompts, a measure of distance between some prompted condition and the distribution of conditions in the dataset. Sub-figure (d) shows a generated story for a novel molecule generated by MolMiner (read left to right, top to bottom)

generation process is transparent, which allows human intervention and the possibility of hybrid computer-human co-design of molecules.

In future work we aim to expand the capabilities of this autoregressive approach of molecular stories including other applications like molecular graph to graph translation and synthesis path prediction.

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

# A   APPENDIX

## A.1   EXAMPLE OF FRAGMENT EXTRACTION

Example of the fragments extracted using Algorithm 1 on the molecule with SMILES
'O=C1CC(=O)C=C1C(=O)O'.

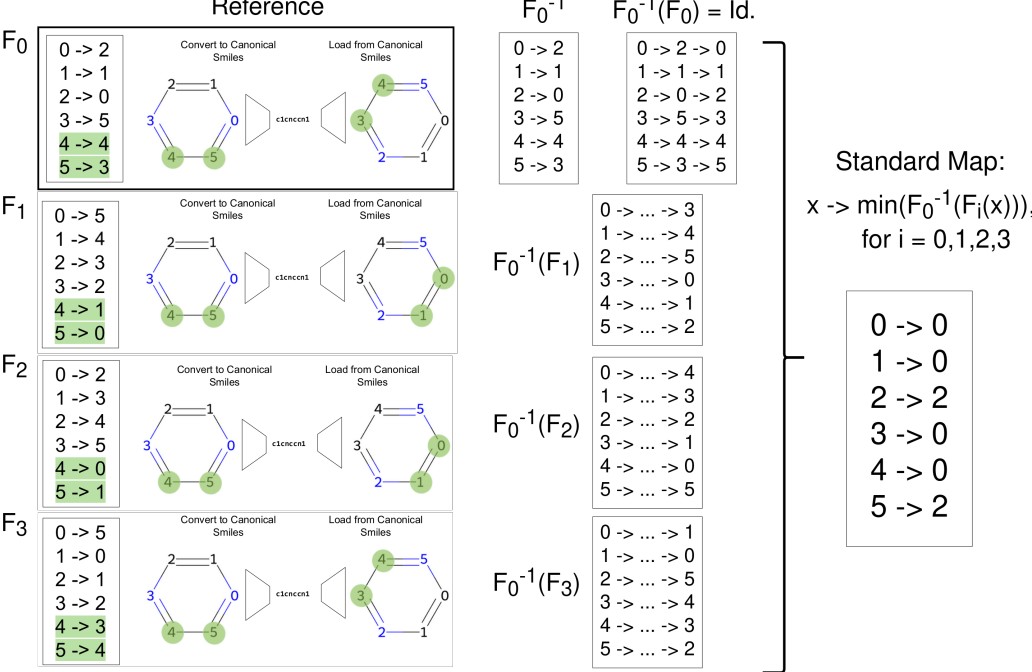

Figure 6: Extracted fragments highlighted on the molecular graph for
'O=C1CC(=O)C=C1C(=O)O'

## A.2   CONSTRUCTION OF THE STANDARD MAP

Figure 7 illustrates the process of creating the standard map, used to uniquely identify attachment
points taking into account symmetries. Given all possible maps from one indexing to another, rep-
resented as $F_i$ for $i = 0, 1, 2, 3$ the standard map is constructed by taking the first map as reference,
$F_0$, inverting it and convoluting it with all other maps, then taking the minimum value mapped to
$x \rightarrow min(F_0^{-1}(F_i(x)))$.

Figure 7: Extraction of fragments and attachments from a molecule. Construction of the standard
map.

### A.3 EFFECT OF THE NUMBER OF TOP-K FRAGMENTS IN MOLECULAR GENERATION

When generating a molecule using MolMiner we start by selecting a starting fragment from which the whole molecule will be grown. A model was trained to predict probability of any fragment manifesting, given the target conditions. When we are generating a new molecule we only need one starting fragment, so we can greedily take the one with the highest predicted probability of manifesting and use it. One problem of this greedy approach is the loss of diversity. In the calibration experiment, the same prompts where used to create 30 molecules, so if using only the top-1 starting fragments the model would always be using the same starting fragment. In Figure 8 we explored the effect of using the top-1, top-3 or top-5 starting fragments in the calibration of the Synthetic Accessibility score. The three different options perform equally for lower values of SAScore. The top-1 performs noticeably worse for higher values, with its mean departing the ideal correlation before top-3 or top-5 do, and the standard deviation is greater for values of SAScore greater then 3.0 than the two other options. Top-3 and top-5 perform similarly in the entire range, with top-3 performing slightly better for SAScore values in (1.5,2.0) and (3.5,4.0), which motivated its used in the main text.

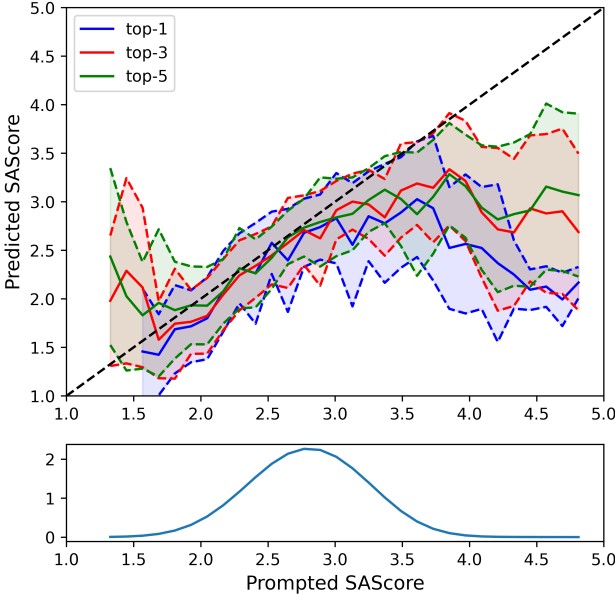

Figure 8: Calibration for SAScore using top-1, top-3 and top-5 starting fragments.

### A.4 THE ROLE OF INCORPORATING GEOMETRY

Using the optimized network parameters from the hyperparameter search, we explored the effect of different initialization methods for the weight on the geometry in the modified attention mechanism. Using the same 100 molecules from the training set and 100 molecules of the validation set 10 models were trained for 100 epochs, or for 10000 stories. Each of the different models had a different initialization method for the scalar parameter. Including gaussian, uniform and constant initialization. One particular initialization is interesting for its implications: choosing the initial value as constant 0.0 and not letting the model optimize this parameter results in complete removal of geometric information of the model. The metrics monitored are the classification accuracy for both the training and validation set. The results are shown in Table 2. The accuracy metrics are approximately the same for all the different models, which we believe indicates the network is plastic enough that it can learn equally well to fit this reduced data regardless of the initialization method. However upon closer inspection, one can see that models with constant and positive initialization perform sightlier better, in particular models with Constant=+1.0 achieves the best accuracy on the validation set and Constant=+10.0 does so on the training set. This is in line with what one might think, the positive sign means the attention to further away fragments is reduced, whereas a neg-

Table 2: Ablation study of the scalar weight of the geometry. Reduced dataset

| Initialization | Training accuracy (%) | Validation accuracy (%) |
|---|---|---|
| Normal(0,1) | 80.37 | 79.90 |
| Normal(1,1) | 79.95 | 79.30 |
| Uniform(-1,1) | 79.98 | 79.41 |
| Uniform(0,1) | 79.85 | 79.48 |
| Constant=-10.0 | 80.14 | 79.99 |
| Constant=-1.0 | 80.01 | 80.06 |
| No Geometry | 80.37 | 79.84 |
| Constant=+0.1 | 80.11 | 80.02 |
| Constant=+1.0 | 80.93 | 80.09 |
| Constant=+10.0 | 81.25 | 79.41 |

Table 3: Effect of the geometry. Entire dataset.

| Initialization | Training accuracy (%) | Testing accuracy (%) |
|---|---|---|
| No Geometry | 80.58 | 80.31 |
| Constant=+1.0 | 81.98 | 81.17 |

ative sign would do the opposite and would be counter-intuitive. It is worth noting also that the No-Geometry model performed relatively good.

However the difference in this small validation set is so small its not fair to draw conclusions from it. In order to finally asses the role of the geometry, two different models are trained on the entire dataset one with Constant=+1.0 since it performed best at the validation set, and a model without Geometry. The results of this last experiment are shown in Table 3. The model with geometry outperforms the model without geometry by approximately 1 point in both training and testing set. Further work will be carried out to explore the dependence on datasets of geometry aware vs geometry not aware models, as this is a widespread problem within machine learning for materials (Tian et al., 2022).

### A.5 COMPARISON ON FRAGMENTATION APPROACHES

The fragment extraction procedure described in Algorithm 1 found the 34 fragments shown in Figure 9. As discussed in the main text the motivation behind our particular choice of fragmentation was based on our desire for the fragments to be unique, disjoint and interpretable. We then showed that our particular procedure results in fragments that are always single-cyclic graphs which allowed us to handle their symmetries and avoid the presence of duplicate entries in the vocabulary formed by the tuple (fragment, attachment).

In this subsection we compare our approach with that of previous work and highlight some unreported problematic results of previous methods, that in turn will emphasize the importance of our contribution.

Our fragmentation approach, Algorithm 1, is the simplest procedure of extracting fragments when compared to the approach used in JTNN(Jin et al., 2019), HierVAE(Jin et al., 2020) or HierDiff (Qiang et al., 2023). It must be noted that this three other approaches are not independent but are closely related. Both HierDiff and HierVAE are based on JTNN's fragmentation. All these methods, calculate the Small Set of Small Rings (SSSR) of the molecule, which they use to divide the molecule into ts constituent rings, and bonds. Then, HierVAE, HierDiff and JTNN merge individual rings if they share two or more atoms, and HierVAE also re-merges all bonds mutually connected into a single fragment. Algorithm 1 does not do any further merging, once the rings and bonds are extracted they are not fused together, and the result of this is a set of fragments that are single-cyclic graphs, as discussed in the main text.

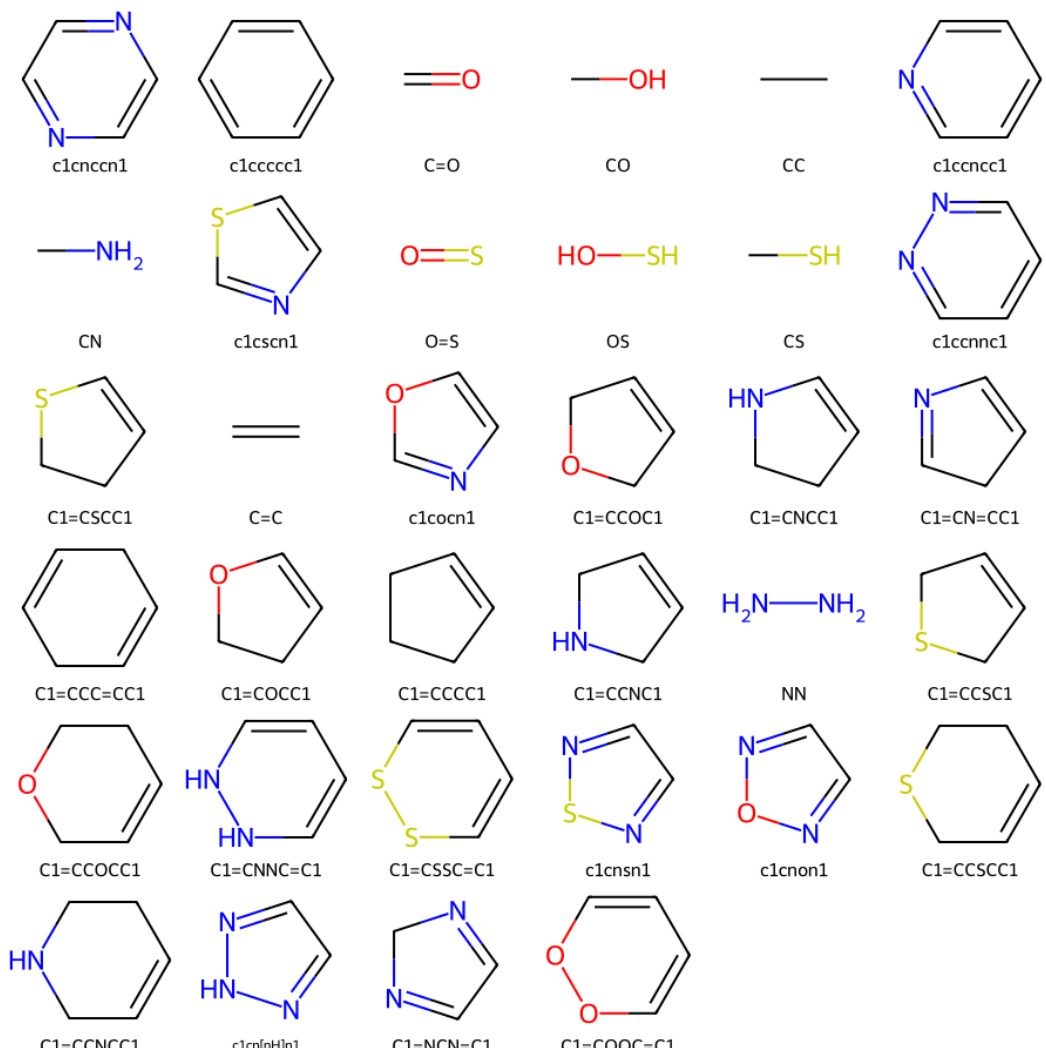

Figure 9: Fragments (without attachments) extracted for RedDB dataset (Sorkun et al., 2022). Using Algorithm 1.

The problem of HierDiff, JTNN and HierVAE handling of the fragments can be highlighted with an example. Looking at the vocabulary of fragments of HierVAEs implementation, which can be found at https://github.com/wengong-jin/hgraph2graph/blob/master/data/logp04/vocab.txt lines 30-40 correspond to different attachment configurations of the fragment with SMILES C1=CC=[NH+]C=C1. We note that among the different attachments of the fragment there are duplicates: a): C1=C[CH:1]=CC=[NH+]1 and b): C1=[NH+]C=C[CH:1]=C1 are the same attachment configuration (Note this is indicated by a number next to the atom e.g C:1), but its only because their model does not take into account the symmetries that this is possible. Now, if during training, the model predicts a.) but the *correct answer* is b) that would prevent the model to learn correctly. Our model deals with this by incorporating the symmetries. A visualization of the duplicates attachments in the vocabulary of fragments of HierVAE is shown in Figure 10.

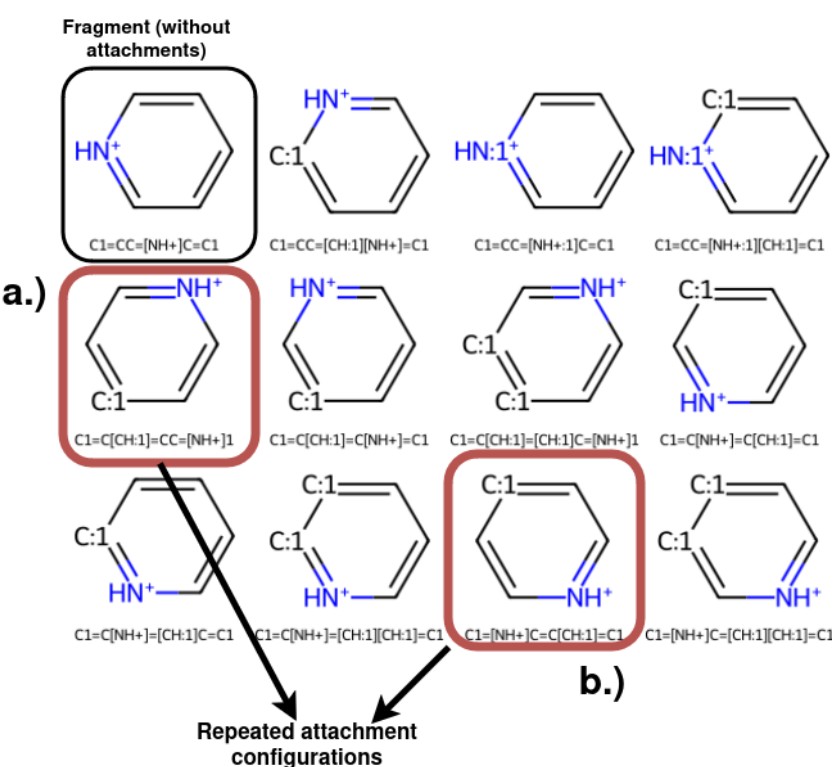

Figure 10: Duplicated vocabulary entries in HierVae.

### A.6  ON THE IMPORTANCE OF VARIABLE-SIZE AUTOREGRESSIVE GENERATION OF MOLECULES

Enabling variable size of the molecule during generation signifies the removal of an arbitrary constraint. Why should any model pre-define before generation the size (in terms of atoms or in terms of fragments) of a molecule? We believe this is a result of adapting generative approaches from Vision tasks (image generation, where the image has a fixed size) to materials modelling without taking into account the unique challenges of materials. In this subsection we would like to raise awareness of the limitations of fixing the size of the molecule during generation, an approach that is widely extended in current diffusion models for molecules, with examples in (Hoogeboom et al., 2022b), (Qiang et al., 2023).

Given a set of target properties, the space of possible molecules that satisfy those properties do not necessarily have the same size, therefore, by constraining the space of possible solutions, to those with some predetermined size constrains the solution. Given that this is an unnecessary constraint (other than convenience), a justification should be demanded from models that impose it, and not of models that remove it.

To further highlight the error that fixing the size of the molecule upon generation constitutes, we can examine how the size of the molecule is chosen upon initialization in models that pre-define it. From (Qiang et al., 2023) section *C. Additional Experiments* subsection *C.1. Experimental configuration.* *'In all non-autoregressive methods, the number of nodes used for sampling is drawn from the size distribution histogram calculated on the training set.'*. In other words, they sample the size of the molecule from the distribution of the molecular sizes of the dataset. Doing so for non-conditional generation imposes an unjustified constraint as discussed before. However if this is done for conditional generation (section *C.6. Conditional generation* from (Qiang et al., 2023)) then this is altogether wrong. Because they are sampling from the distribution of sizes when they should be sampling from the conditional distribution of size given the property. In other words, in general given a set of target properties the distribution of sizes of molecules satisfying those properties is not the general distribution of sizes.

### A.7  BENCHMARKING MOLMINER: CALIBRATION, NOVELTY AND SAMPLE EFFICIENCY

We aim to compare the performance of our proposed model, Molminer, against its closest relative in terms of the modeling approach, HierVAE (Jin et al., 2020). We believe that the best way to compare these two models is by evaluating them on the task of conditional generation of novel molecules. That is, how good are these models at generating novel molecules that obey desired target criteria?.

To achieve so, we need to modify HierVAE to allow for conditional generation. HierVAE is a latent-space based generative model, so we can concatenate conditions to the latent vector and decode using this extended latent space. After implementing this slight modification, both HierVAE and Molminer were trained on the same training set for the same number of epochs, 50. Once trained we use them to generated compounds according to a range of conditions of Solubility, Synthetic Accessibility and Redox Potential spanning the whole dataset, as was done in the Calibration experiment, resulting in three separate experiments. The result of this experiment is shown in Figure 11 and Table 4

Table 4: Novelty percentage of the total generations for each of the three experiments, for each model. Note that Molminer generates a significantly higher percentage of novel molecules than HierVAE.

| Model | SAScore-exp | LogSolubility-exp | Redox Potential-exp |
|---|---|---|---|
| HierVAE-50epochs | 57.68 % | 58.87 % | 38.57 % |
| MolMiner-50epochs | 81.18 % | 76.60 % | 74.34 % |

In Figure 11 we can see the calibration plot for novel molecules from the three experiments by HierVAE and Molminer after 50 epochs of training. HierVAE is better calibrated than MolMiner, specially for the extreme values for each of the conditions, however MolMiner achieves significantly better novelty ratio, upwards of 74% novel molecules generated by MolMiner vs lower than 59%

for HierVAE. However we suspect that HierVAE appears to be better calibrated than MolMiner at 50 epochs because MolMiner requires from more repetitions to achieve the same calibration as HierVAE, as a result from learning from different stories every time, that is, we hypothesize that MolMiner is less sample-efficient. To test this hypothesis we repeat this benchmarking experiment at a later checkpoint for both models, at 80 epochs. The hypothesis is that, because MolMiner uses different stories at every epoch it takes more epochs to converge, so if we compare these two models at a later training checkpoint the difference in calibration will be lower. This second calibration experiment is shown in Figure 5 and Table 1.

From Figure 5 we can see that both models obtain now similar calibration results, which is compatible with the hypothesis that MolMiner takes more epochs to achieve the same calibration result than HierVAE, most probably due to seeing a different story at evert epoch for every molecule. Again comparing the novelty ratios from Table 1 we obtain similar results as for the 50-epoch experiment, Molminer obtains a novelty ratio upwards of 79 % while HierVAE obtains a novelty ratio lower than 56 %.

MolMiner can achieve controlled generation of novel molecule at a better ratio than its predecessor. However we note that the model is less sample-efficient, requiring from more epochs to achieve the same performance than its predecessor due to the fact that MolMiner sees a different story for every molecule at every epoch.

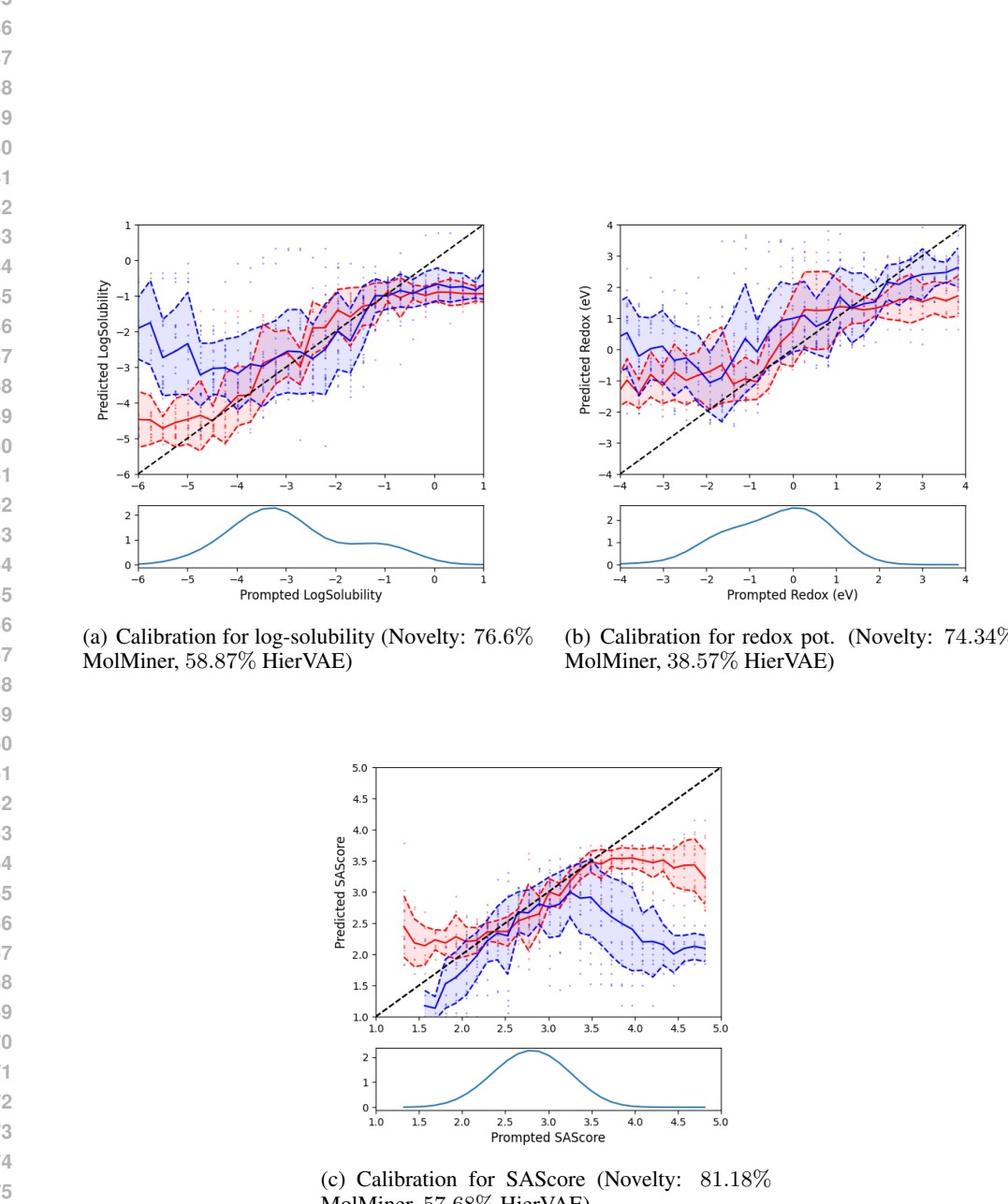

(a) Calibration for log-solubility (Novelty: 76.6% MolMiner, 58.87% HierVAE)

(b) Calibration for redox pot. (Novelty: 74.34% MolMiner, 38.57% HierVAE)

(c) Calibration for SAScore (Novelty: 81.18% MolMiner, 57.68% HierVAE)

Figure 11: Calibration plots (red: HierVAE-50epochs, blue: MolMiner-50epochs) prompted vs predicted condition for the three properties trained on: (a) log-solubility, (b) redox potential, and (c) SAScore. The lower half of the figures represents the density of the KDE of each of the prompts, a measure of distance between some prompted condition and the distribution of conditions in the dataset.

## A.8 EXAMPLES OF GENERATED STORIES FOR MOLECULE NOT IN THE DATASET

The figures below show some examples of stories generated for molecules not found in the dataset. These molecule were generated during the calibration experiments. Note that the cauterization steps are omitted in the visualization to avoid repetition.

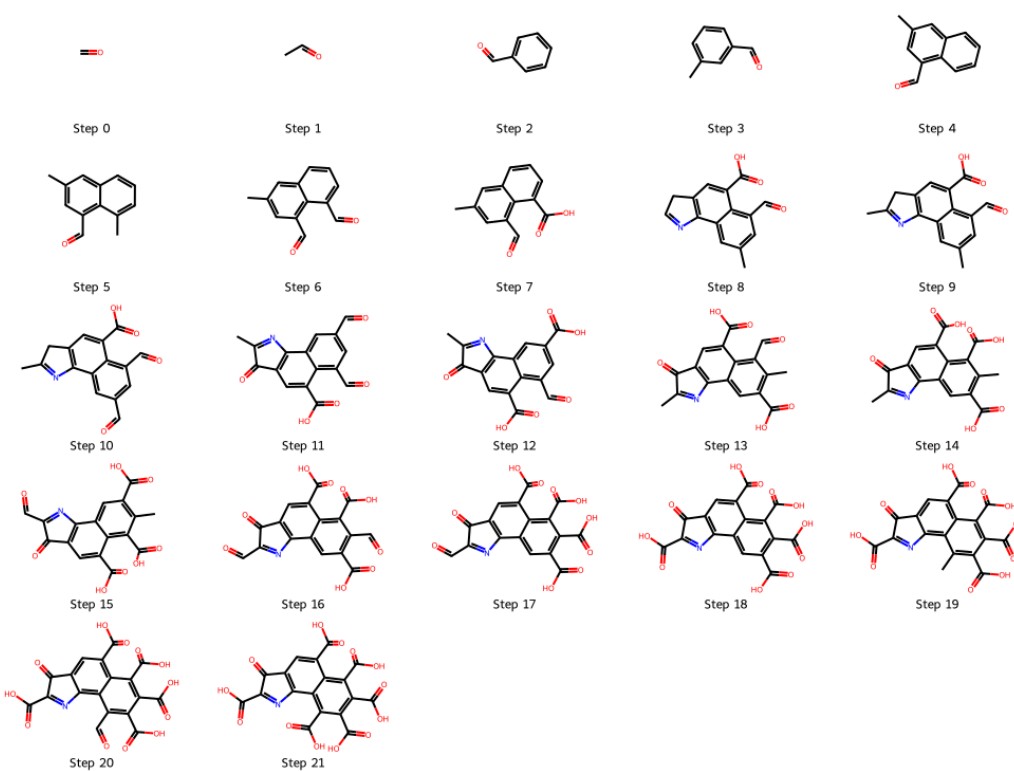

Figure 12: Example of story generated in the creation of a compound not in the dataset.

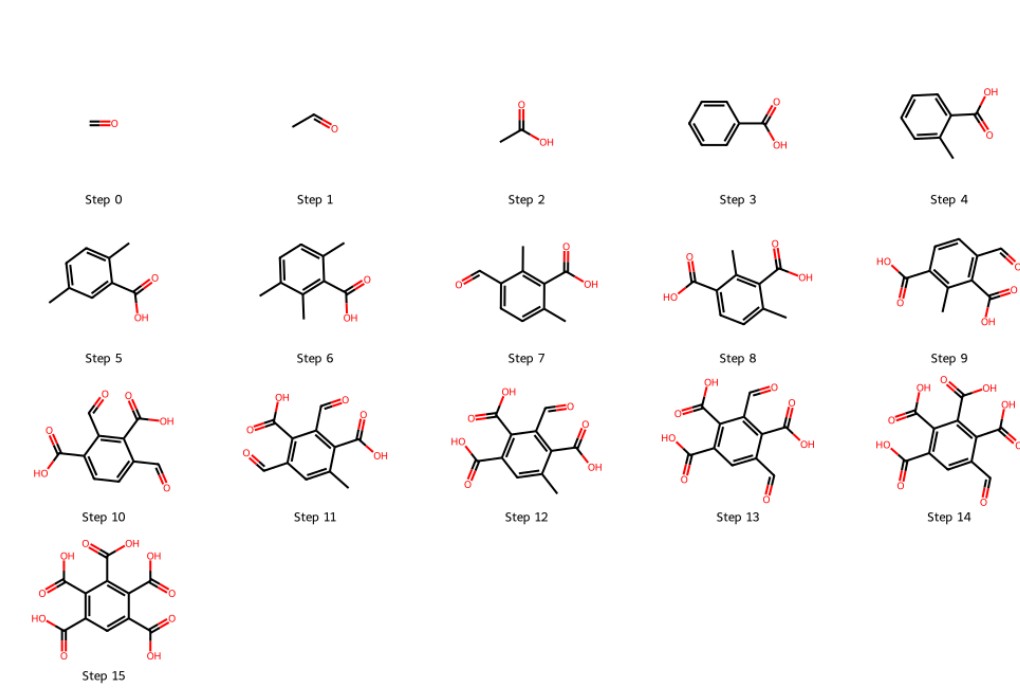

Figure 13: Example of story generated in the creation of a compound not in the dataset.

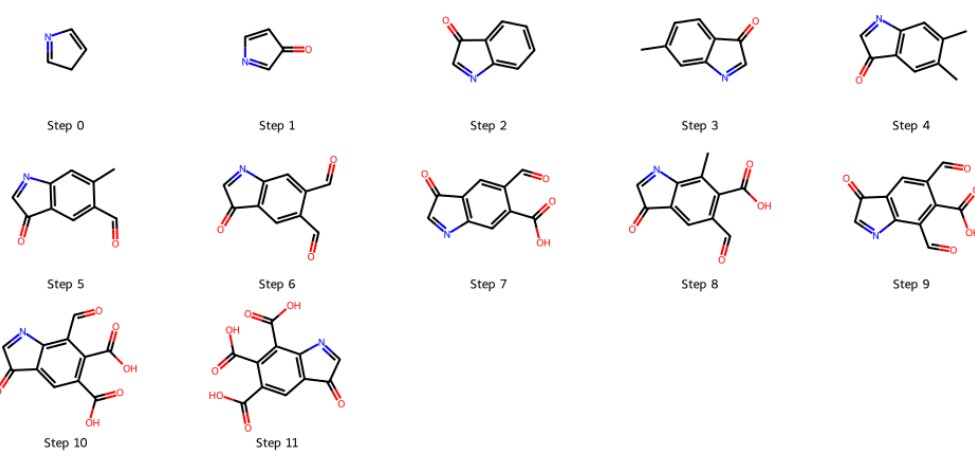

Figure 14: Example of story generated in the creation of a compound not in the dataset.

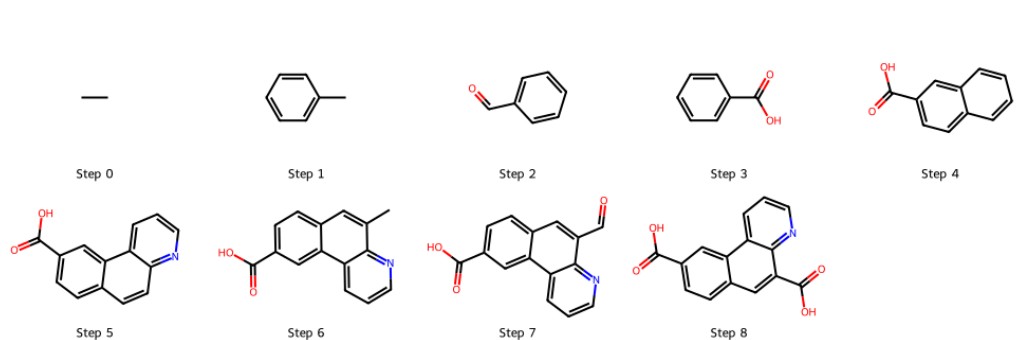

Figure 15: Example of story generated in the creation of a compound not in the dataset.

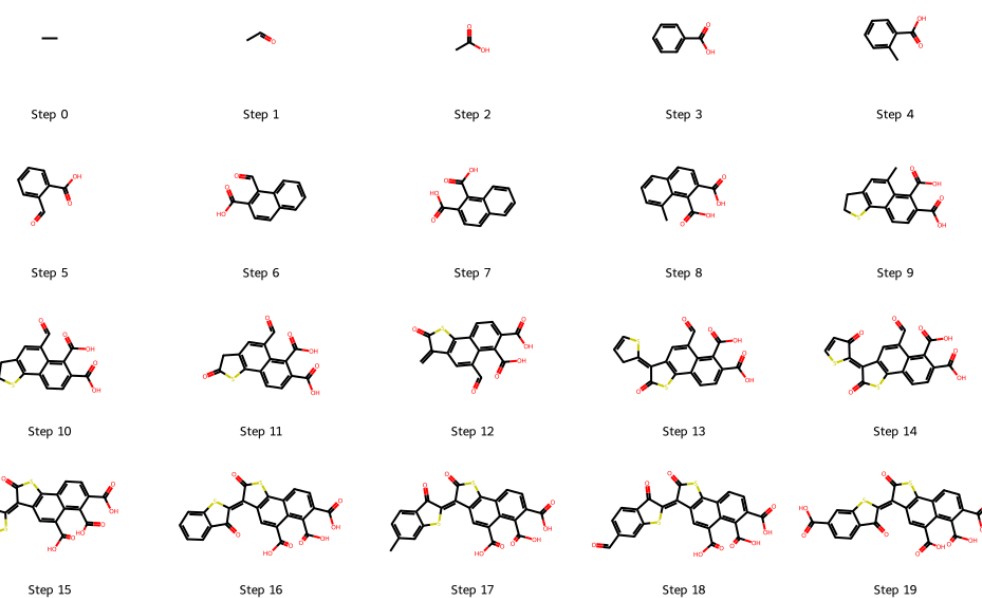

Figure 16: Example of story generated in the creation of a compound not in the dataset.

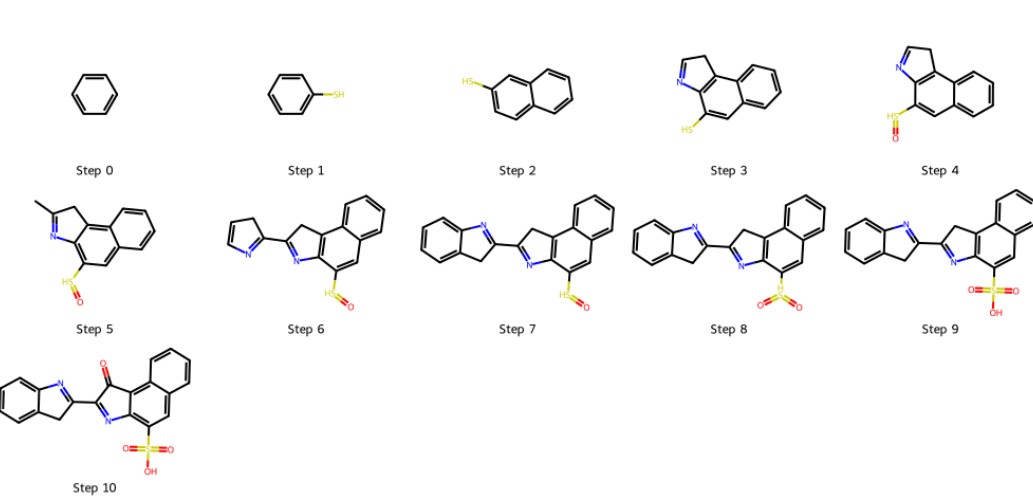

Figure 17: Example of story generated in the creation of a compound not in the dataset.

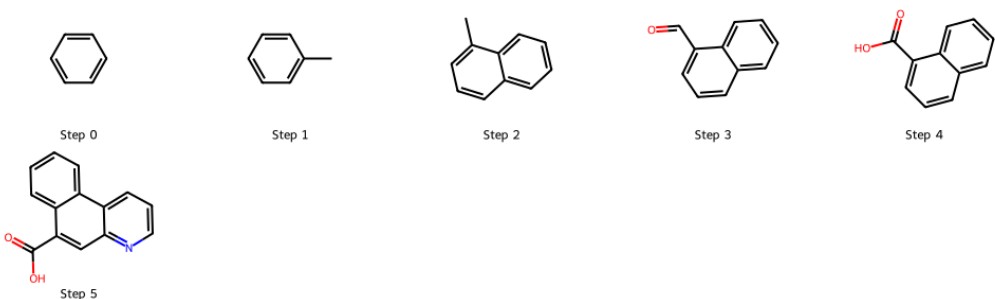

Figure 18: Example of story generated in the creation of a compound not in the dataset.

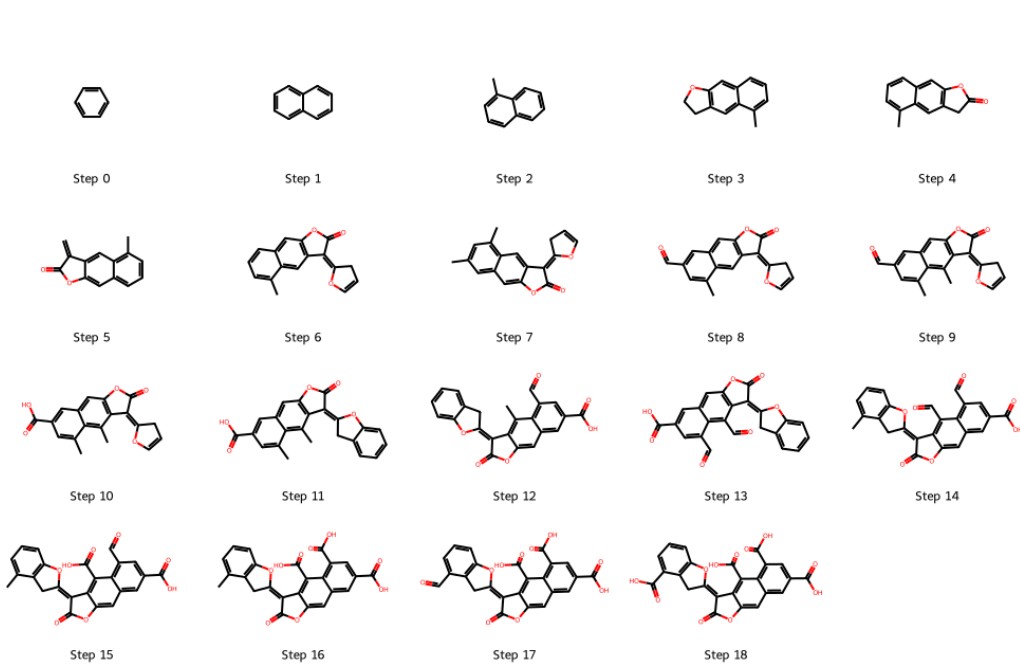

Figure 19: Example of story generated in the creation of a compound not in the dataset.

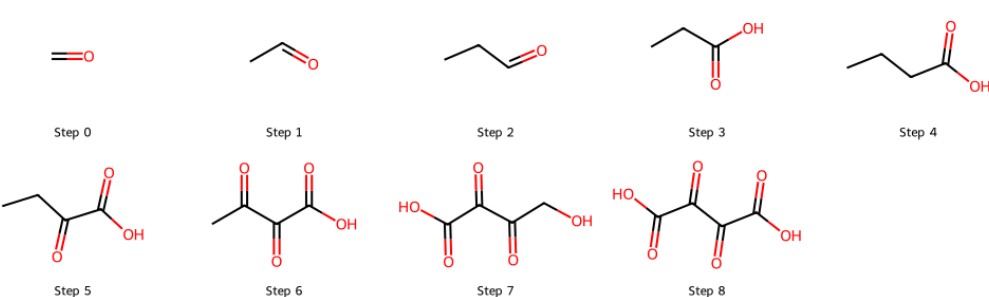

Figure 20: Example of story generated in the creation of a compound not in the dataset.

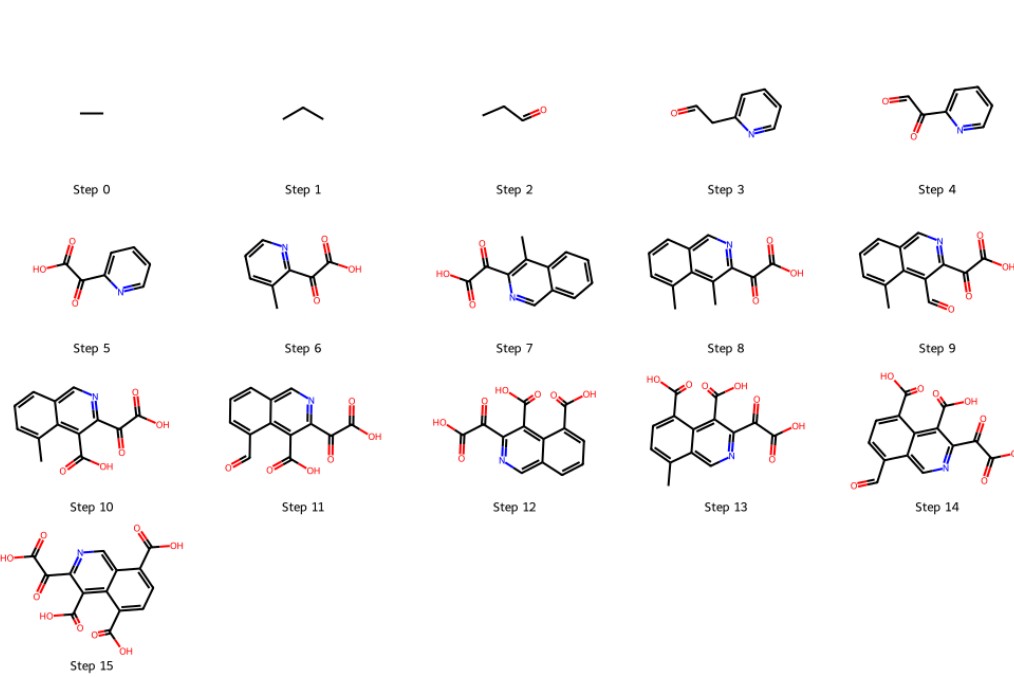

Figure 21: Example of story generated in the creation of a compound not in the dataset.

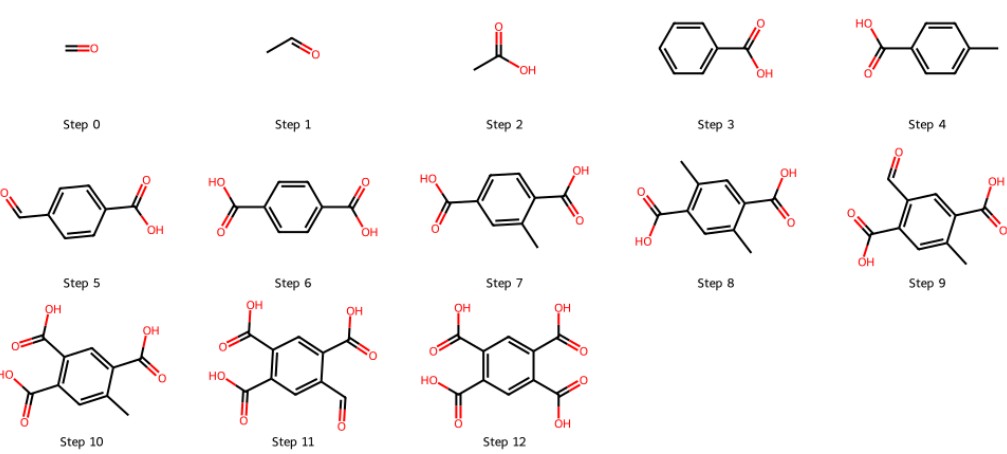

Figure 22: Example of story generated in the creation of a compound not in the dataset.

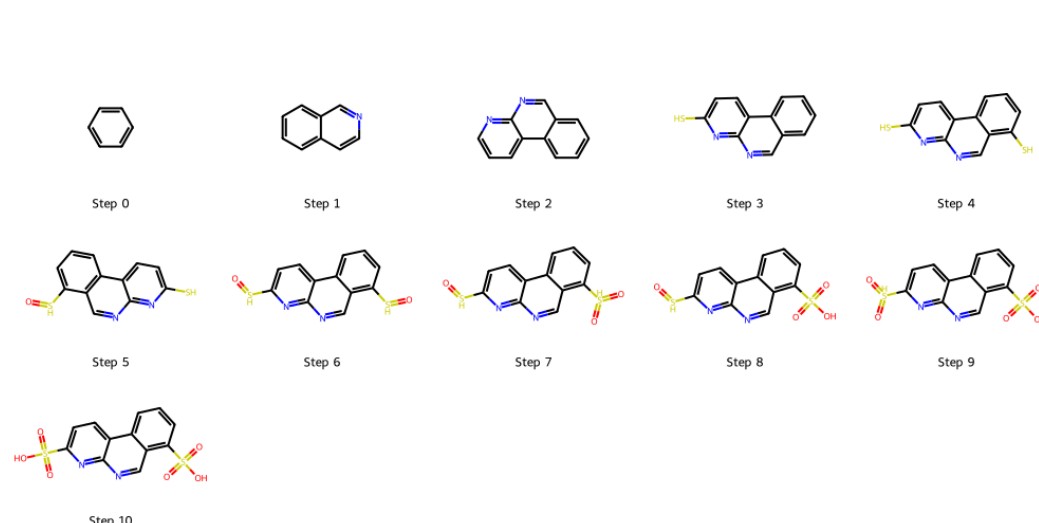

Figure 23: Example of story generated in the creation of a compound not in the dataset.

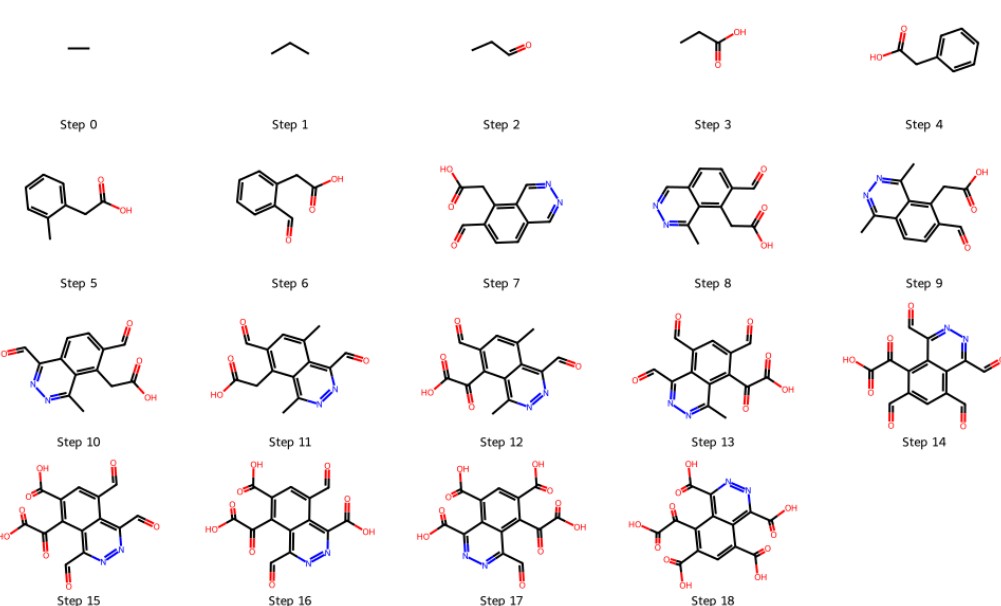

Figure 24: Example of story generated in the creation of a compound not in the dataset.

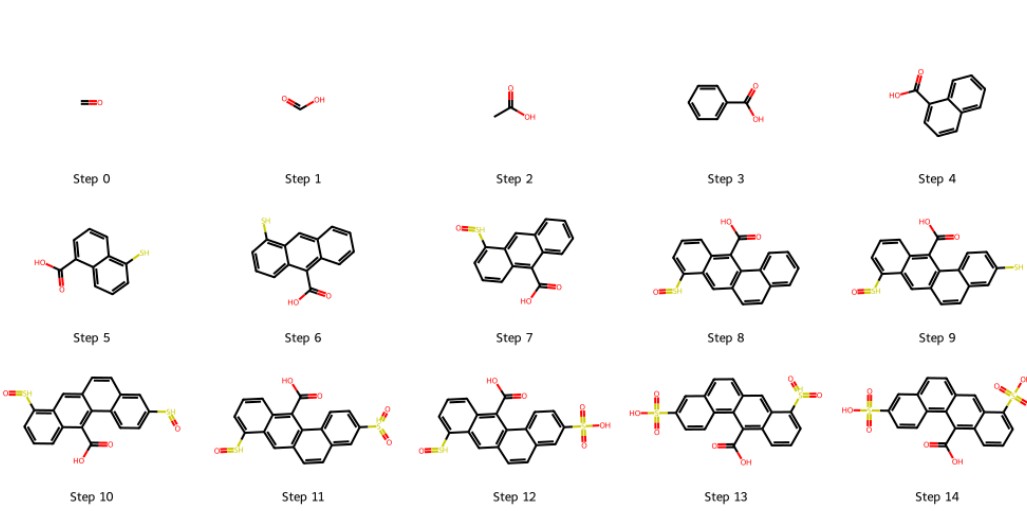

Figure 25: Example of story generated in the creation of a compound not in the dataset.

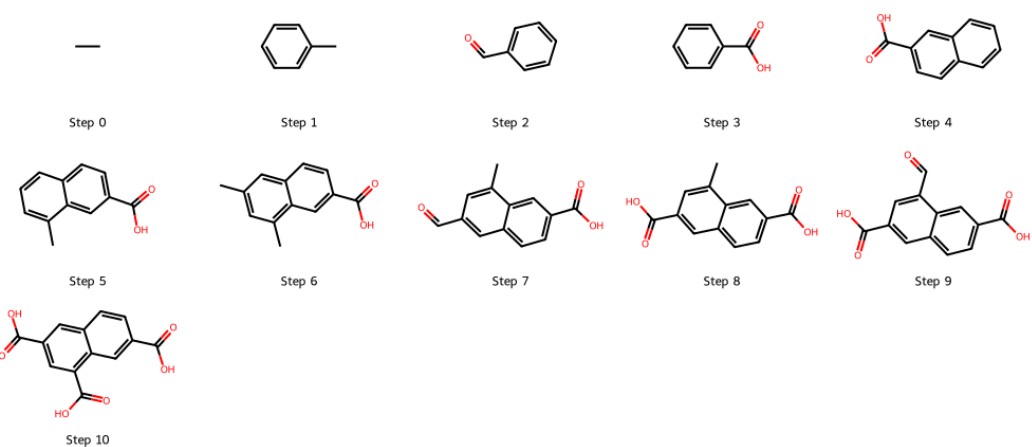

Figure 26: Example of story generated in the creation of a compound not in the dataset.

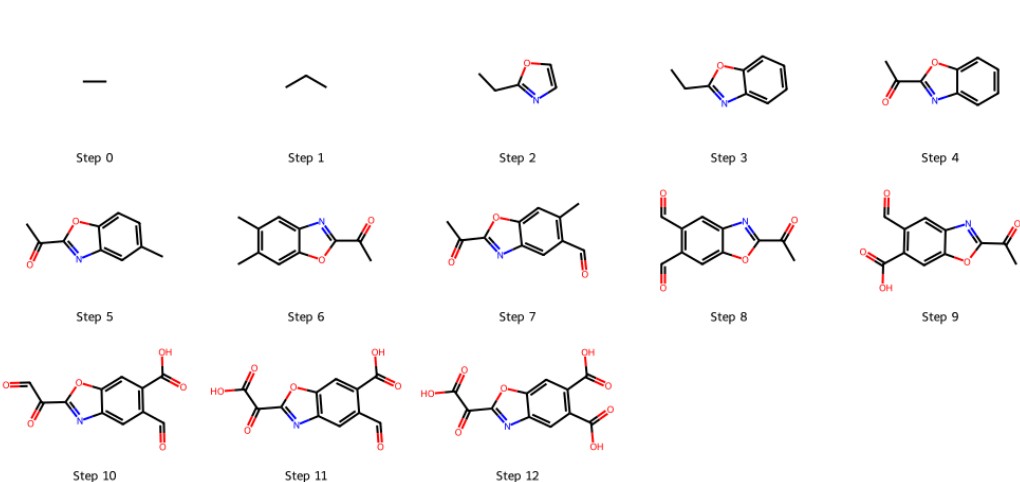

Figure 27: Example of story generated in the creation of a compound not in the dataset.

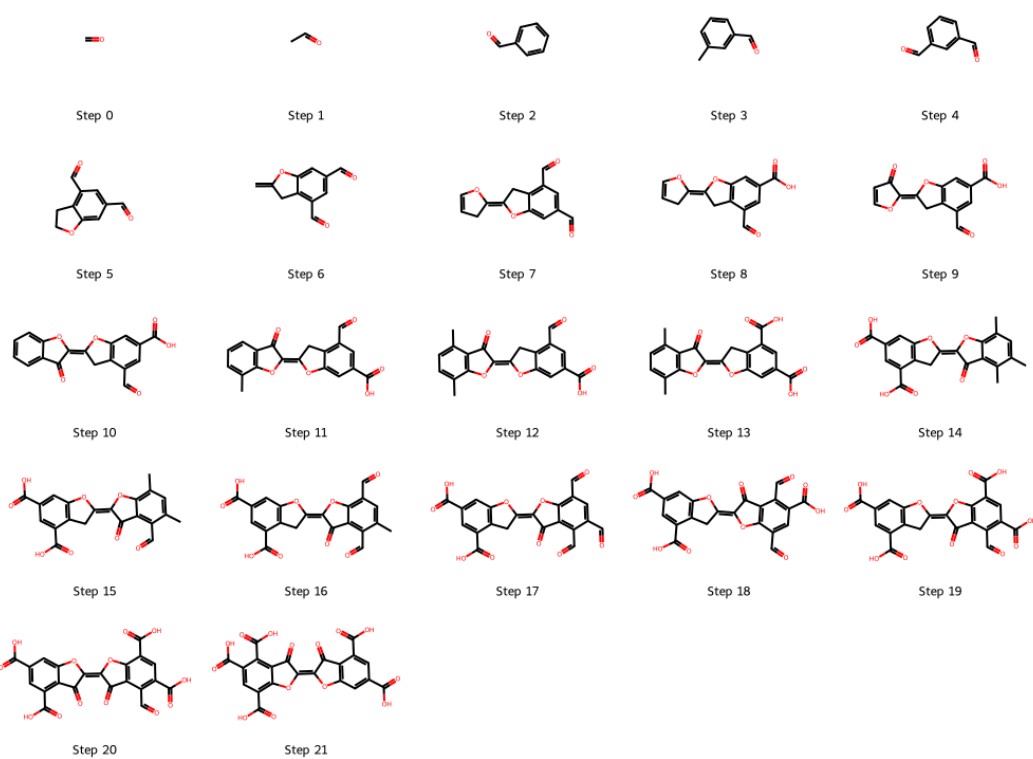

Figure 28: Example of story generated in the creation of a compound not in the dataset.

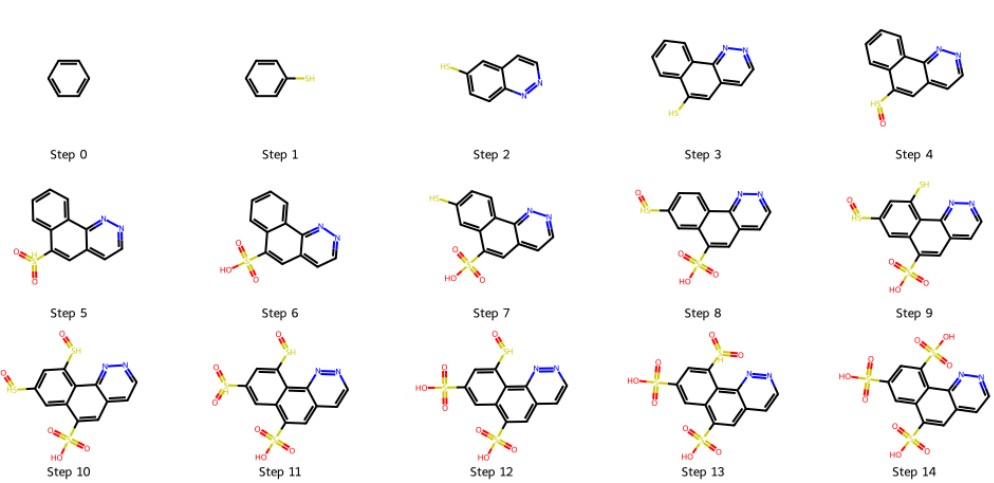

Figure 29: Example of story generated in the creation of a compound not in the dataset.

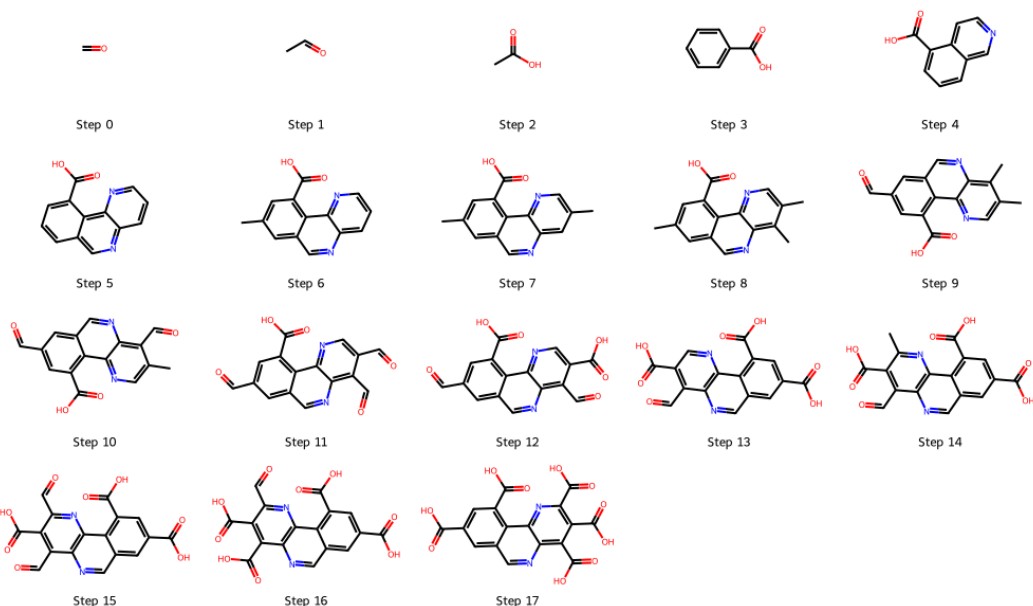

Figure 30: Example of story generated in the creation of a compound not in the dataset.

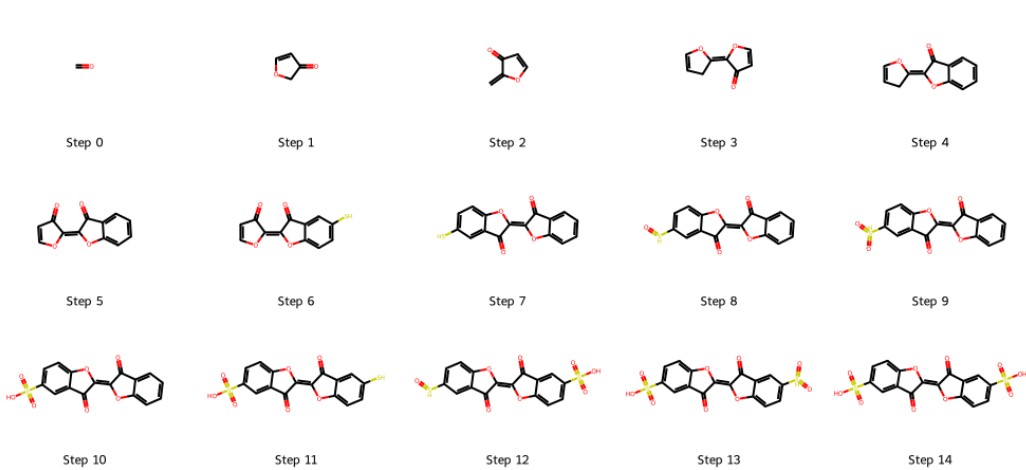

Figure 31: Example of story generated in the creation of a compound not in the dataset.

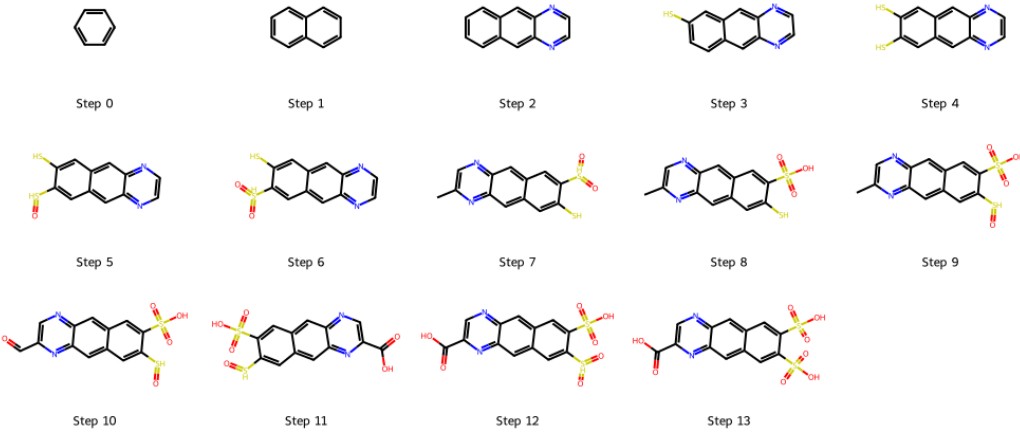

Figure 32: Example of story generated in the creation of a compound not in the dataset.

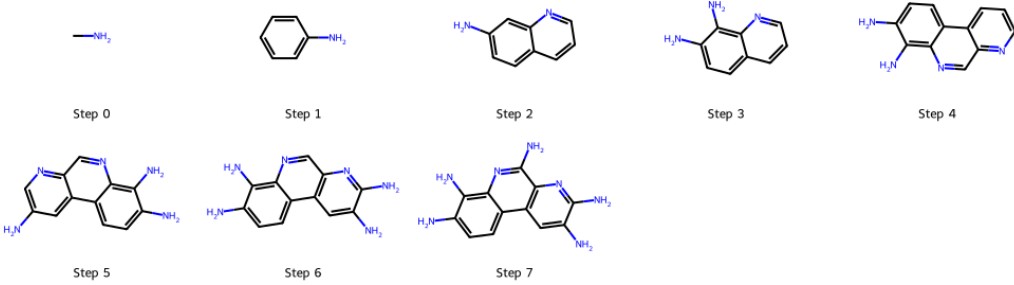

Figure 33: Example of story generated in the creation of a compound not in the dataset.

