# OpenReview forum: "MolMiner: Transformer architecture for fragment-based autoregressive generation of molecular stories"
_ICLR.cc/2025/Conference — Submitted to ICLR 2025_

### Official Review · Reviewer_YR4n · 2024-11-02

**Soundness:** 1
**Presentation:** 2
**Contribution:** 1
**Rating:** 3
**Confidence:** 4

**Summary:**

This paper presents MolMiner, a transformer-based autoregressive model to generate molecules by assembling fragments. This paper details on fragment pre-processing, specifically on how it assigns indices to atoms in fragments and canonicalizes indices of attach points (by using the minimum index in symmetrical cases). A decoder-only transformer network is used to generate a sequence of (fragment, attach point) tuples to build molecular graphs.

**Strengths:**

- Generating molecules by fragments rather than by atoms is more likely to create reasonable molecular structures and hence increases validity.
- Detailed explanation on fragment preprocessing and atom indexing.

**Weaknesses:**

- The core idea of this paper, generating molecules by motifs or fragments, has been practiced for very long time. There have been review articles discussing the progress in this area [1, 2]. This paper overlooked the progress in this area and in the current context the novelty of the proposed method is very limited.
- The evalution is very weak. This paper only demonstrates MolMiner's application in **only** three properties, specifically, SAScore, solubility, and redox potential. Evaluting the model solely on them is not a strong evidence of praticality of the proposed models, as these scores are somehow week and simple.
- The Introduction section mentioned "Incorporating 3D information". However, I did not notice how 3D information was significantly incorporated in both method formulation and evaluation.
- The concept of "molecular story" is weird. A "molecular story" is basically an auto-regressive generation process of a sequence of molecular fragments. There seems to be nothing else that makes the generation process unique or novel such that it needs a new name to describe itself. It seems to me that the "molecular story" name leads to confusion and ambiguity.

[1] Sergei Voloboev. A Review on Fragment-based De Novo 2D Molecule Generation (2024)

[2] Shao Jinsong et al. Molecular fragmentation as a crucial step in the AI-based drug development pathway (2024)

**Questions:**

See Weaknesses

---

> ### Author Response · Authors · 2024-11-19
> **Weaknesses 1,  2 & 3**
>
> Thanks a lot for the review. We appreciate the comments and the criticism. However we believe there has been some misunderstandings, and we hope these issues will be clarified with our revised manuscript and the following answers. We address here the weaknesses pointed out by the reviewer, in the same order.
>
> # Weakness 1:
> While indeed people have worked on generating molecules by fragments, it has not been solved, this is clearly highlighted in the conclusions of the review articles the reviewer points out [1], from their conclusions:
> > “Looking forward, we identify these research directions as promising for the field:
> Transparency and Uncertainty Quantification. Understanding how models decide on specific fragment is crucial for critical areas such as drug design (Deng et al., 2021).Integrating uncertainty quantification into models can identify reliable molecules with promising properties, thereby streamlining the drug design process (Hu et al., 2023).
> Focus on Realistic Properties. While many studies focus on computational metrics such as logP or QED11, there is a need for experimental validation of lead molecules. Future research could shift towards the generation of molecules optimized for properties that are realistically testable and applicable (Bilodeau et al., 2022).
> Advancing 3D Molecule Generation. The transition to 3D molecular representations is critical for structure-specific tasks. The application of fragment-based methods to 3D molecule generation provides an opportunity to address validity challenges in this area. Despite initial studies (Qiang et al., 2023), further exploration in this area is needed”
>
> In light of this review paper [1] the reviewer points out, our model is clearly novel. In view of the highlighted points in the conclusion from [1] we can clearly see how our model addresses points 2.) and 3.) directly:
> - We use properties like redox-potential, log-solubility and synthetic accessibility which are real world properties and key for the design of electroactive organic molecules for aqueous redox flow batteries, a promising alternative for energy storage. Moreover we evaluate our model on a significantly harder dataset than QM9, since QM9 contains molecules up to 9 heavy atoms, while the dataset we used contains molecules of up to 54 heavy atoms (mean of 24). This constitutes a significantly more realistic scenario. And we optimize the three properties simultaneously.
> - We propose a viable solution for integrating 3D coordinates into an autoregressive model. Directly addressing a notable challenge of this type of models.
>
> In view of the evident connection of our contributions to the needs and ‘research directions’ outlined in the review paper [1] pointed out by the reviewer we are more confident than our contributions are significant. We have now added the review paper to further contextualize the importance and contributions of our work.
>
> # Weakness 2
> >The evalution is very weak. This paper only demonstrates MolMiner's application in only three properties, specifically, SAScore, solubility, and redox potential. Evaluting the model solely on them is not a strong evidence of praticality of the proposed models, as these scores are somehow week and simple.
>
> _Note: We have now performed a benchmark comparison against a well established autoregressive model of the same characteristics, this is now added to the main test. We believe this further strengths our evaluation._
>
> Demonstrating simultaneous multi property conditional generation using three properties, and in particular our choice of realistic and useful properties for the design of organic molecules for aqueous redox flow batteries is not a minor challenge in the least. We completely disagree that Redox potential or Solubility are ‘weak’ and ‘simple’ properties to do conditional inverse design. These are properties that are hard to predict, even with traditional simulations like DFT, which is among  the reasons why finding suitable organic compounds for this type of batteries remains a challenging problem.
>
>
> # Weakness 3
>
> > The Introduction section mentioned "Incorporating 3D information". However, I did not notice how 3D information was significantly incorporated in both method formulation and evaluation.
>
> Lines 255-258 introduce how the 3D information is computed at every step of a story, lines 296-299, 303-304 introduce how the 3D information is fed into the attention mechanism of the Transformer. Figure 3, on the right, Inside the architecture of the transformer, the specific point at which the geometry is introduced into the network is colored in red, the process is also described in the caption of figure 3. In figure 4 the pairwise-distance matrix is highlighted in red, with the caption ‘Pairwise distances between fragments. Then Apendix “A.4 The role of the geometry” (lines 795-841) discuss the effect of the geometry and point into future work regarding the role inclusion of geometry in ML models.

---

> > ### Author Response · Authors · 2024-11-19
> > **Weakness 4**
> >
> > # Weakness 4
> >
> > > The concept of "molecular story" is weird. A "molecular story" is basically an auto-regressive generation process of a sequence of molecular fragments. There seems to be nothing else that makes the generation process unique or novel such that it needs a new name to describe itself. It seems to me that the "molecular story" name leads to confusion and ambiguity.
> >
> > A ‘molecular story’ is indeed one particular generation sequence of a molecule. We called it a story to emphasize that some molecule can have many possible generations or stories. This is similar to the term ‘rollouts’ in reinforcement learning, or in general, a sequence. We called it a story to differentiate that is not just a sequence, is a sequence with a particular set of properties: All the intermediate steps are valid. We could have called it trajectory, but this would be confusing with molecular dynamics usage of ‘trajectory’.

---

### Official Review · Reviewer_MQYH · 2024-11-04

**Soundness:** 3
**Presentation:** 3
**Contribution:** 3
**Rating:** 6
**Confidence:** 3

**Summary:**

This paper introduces MolMiner, a novel generative model designed to tackle challenges in molecular generation by leveraging a fragment-based, autoregressive approach. The core idea is to generate molecules step-by-step using molecular fragments, which makes the generation process more interpretable and allows the model to decide the molecule size dynamically.

MolMiner addresses several key challenges in generative molecular modeling, including chemical validity, flexibility in molecular size, and incorporating 3D spatial information. By enforcing chemical rules during the generation process, MolMiner ensures that chemically valid molecules are produced at every step, reducing the risk of invalid outputs. Additionally, the model uses transformer-based geometry-aware mechanisms to account for the spatial arrangement of molecular fragments.

**Strengths:**

The paper's key strength lies in its novel approach to molecular generation, where it effectively combines concepts from natural language processing, such as autoregressive transformers, with the domain-specific requirements of chemistry. By framing molecular synthesis as a sequence of interpretable, fragment-based steps—termed "molecular stories"—the paper addresses significant challenges like chemical validity and lack of transparency in molecular generative models. This reinterpretation of the generative process not only makes the model's decisions easier to understand but also empowers domain experts to validate and interact with the process, bridging an important gap between machine learning and chemistry.

The use of geometry-aware transformers adds another layer of sophistication by ensuring that the model can capture the spatial properties of molecular structures, which are crucial for generating chemically meaningful compounds. The integration of chemical rules during generation to enforce validity is another well-executed innovation that strengthens the quality of the model. These aspects collectively offer a comprehensive and robust approach, improving upon limitations seen in previous one-shot or black-box generative models.

The empirical results are also well-supported, with extensive experiments that validate the model's ability to generate novel molecules aligned with desired properties. This showcases the model's potential for real-world applications in drug and material discovery. Overall, the originality in problem formulation, combined with a thoughtful implementation that directly addresses core limitations in prior models, makes this work a significant and valuable contribution to the field.

**Weaknesses:**

Limited Empirical Scope: The experimental evaluation is restricted to a narrow set of target properties (log-solubility, redox potential, synthetic accessibility). Broader validation, including additional properties (e.g., pharmacokinetics or biological activity) and diverse datasets (such as QM9 or ZINC), would substantiate the versatility of MolMiner.

Lack of Interpretability Quantification: While the model's interpretability is highlighted, the benefits are not explicitly demonstrated or quantified. User studies involving domain experts or real-world case studies would provide practical evidence of the interpretability advantages claimed by the authors.

Fragment Selection Analysis: The chosen fragment extraction method (SSSR) is not thoroughly justified or compared to alternatives. Including an ablation study to explore different fragmentation methods and their impact on model performance would enhance the robustness of the proposed approach.

**Questions:**

See weakness

---

> ### Author Response · Authors · 2024-11-19
> **Weakness 1 & 2**
>
> We would like to thanks the reviewer for its valuable work. We would like to use this comments to address some of the issues raised by the reviewer (most notably we have now benchmarked our model against its predecessor, to further highlight our contribution) and also clarify some points we may disagree on.
>
> # Weakness 1
> > Limited Empirical Scope: The experimental evaluation is restricted to a narrow set of target properties (log-solubility, redox potential, synthetic accessibility). Broader validation, including additional properties (e.g., pharmacokinetics or biological activity) and diverse datasets (such as QM9 or ZINC), would substantiate the versatility of MolMiner.
>
> We disagree that the evaluation on simultaneous multi-property conditional generation using 3 properties constitutes a narrow set, most research done focuses on optimizing one property at a time. In the calibration experiment we perform 3 different experiments, on all of them the 3 properties are simultaneously being used as targets, but on each of the experiments we fix 2 of the properties to the mean while we vary only one at a time. Its important to note, that while we are varying only one, we are optimizing for the three. This is incredibly challenging and we disagree its narrow. The reason why we vary one property at a time, is because otherwise we could not visualize the calibration of simultaneously varying all three properties (that would require a 6-dimensional plot). It must be noted that very often when other work do conditional generation, single or multi-property, they look only at one set of values of the properties, and not to how calibrated the whole model is for all possible values of the single, or multiple properties. In other words, their evaluation, is on a single slice at constant value of the x-axis of our calibration experiments. We evaluate on the whole range of values possible for that target variable.
>
> The set of properties chosen: Synthetic accessibility, redox potential and log-solubility are arguably the key properties to design organic molecules for aqueous redox flow batteries, which is the type of compounds the dataset was initially created for. We had this application in mind because we believe its a promising alternative for energy storage. We didnt include properties like pharmacokinetics or biological activity because the dataset we used did not use those properties in their annotation of molecules, because they are not the most important for the the design of the compounds the dataset is interested in. We showed it works on three arguably challenging properties simultaneously, so we have no reason to believe it would not work on other properties like those mentioned, its just that for the purpose of the dataset, those properties are not a priority.
>
> Finally we did not use QM9 because we wanted to test our model on a more complex, and interesting set of compounds.. QM9 contains molecules with up to 9 heavy atoms, whereas the dataset we used the mean number of heavy atoms in a molecule was 24, with the maximum number of heavy atoms being 54. This constitutes a noticeably harder challenge.
>
> To summarize, we have now benchmarked an existing model against MolMiner on a noticeably challenging task: multi-property simultaneously conditional generation of molecules, training both models in a real-world dataset. We believe this evaluation is challenging and so demonstrates the validity of our approach.
>
> # Weakness 2
>
> > Lack of Interpretability Quantification: While the model's interpretability is highlighted, the benefits are not explicitly demonstrated or quantified. User studies involving domain experts or real-world case studies would provide practical evidence of the interpretability advantages claimed by the authors.
>
> We believe the approach is naturally more interpretable because the generation process is transparent. As can be seen from figures 5d, then figures 10-31 from the appendix, the generation process is decomposed in a sequence of steps. Furthermore, thanks to this, a chemist can intervene at every step of the generation and force a particular choice, then let the model complete the rest. Also one can give the model a partially grown molecule, and tell the model to complete it to satisfy some desired criteria. These possibilities are why we think our model is more transparent or interpretable, than others.

---

> > ### Author Response · Authors · 2024-11-19
> > **Weakness 3**
> >
> > # Weakness 3
> >
> > > Fragment Selection Analysis: The chosen fragment extraction method (SSSR) is not thoroughly justified or compared to alternatives. Including an ablation study to explore different fragmentation methods and their impact on model performance would enhance the robustness of the proposed approach.
> >
> > We disagree, we justify the choice of our particular choice of fragmentation procedure in lines 126-131. Moreover we then further argue the utility of the approach (perhaps less explicitly)  in lines 157-162. There we argue that is thanks to the fact that the fragments extracted using our procedure are always single-cyclic graphs, that we can deal with the symmetries in the way we do. We hope this provides enough justification: First we argue that it satisfies some desirable features ( lines 126-131) then we show that this choice allows us to work with the symmetries in a convenient way ( lines 157-162.).
> >
> > But to further show that our choice of SSSR for fragmenting the molecules is also common, we can take a look at how other work we cite do so.:
> > From HierDiff (https://github.com/qiangbo1222/HierDiff/blob/master/generation/jtnn/chemutils.py#L88C5-L88C51) line 88 we can see Chem.GetSymmSSSR(mol), which is a variant of the SSSR.
> > From HIerVAE (https://github.com/wengong-jin/hgraph2graph/blob/master/hgraph/mol_graph.py#L37) line 37 we can see the same method call Chem.GetSymmSSSR(mol).
> > From JT-VAE (https://github.com/wengong-jin/icml18-jtnn/blob/master/jtnn/chemutils.py#L86) line 86 calls the same method, Chem.GetSymmSSSR(mol).
> > JT-VAE explicitly states the use of this method in the supplementary material under the section ‘A. Tree Decomposition’:_ “Simple rings are extracted via RDKit’s GetSymmSSSR function”_. HierVae or HierDiff do not explicitly state it, but it can be inferred, given that this work is done based on JT-VAE. In any case it can be seen they use the method in their source code. Therefore our use of SSSR we believe is well justified within our work, but also its not uncommon in previous work.

---

> > > ### Comment · Reviewer_MQYH · 2024-11-26
> > >
> > > I appreciate authors for the careful responses and rebuttal.
> > >
> > > After considering the response and other reviewers review. I will keep my current acceptance score.

---

### Official Review · Reviewer_Pmwq · 2024-11-04

**Soundness:** 1
**Presentation:** 2
**Contribution:** 1
**Rating:** 3
**Confidence:** 4

**Summary:**

This paper introduces MolMiner, an autoregressive generative model for molecules that builds them up through fragments. The authors:
* Introduce a technique for deciding on fragments (Section 2.1) -- based on dividing a molecule into its rings and other individual bonds.
* Show how to keep track of fragment attachment points (Section 2.2).
* Describe how they can model the resultant fragment sequence of a molecule (what they call a "molecular story") using an autoregressive model (the model is trained on random orders).
* Show how they can use this model to generate molecules with desirable properties by training a separate feedforward model to decide which fragment to start with. When doing so, MolMiner is able to successfully generate molecules with the desired, targeted property when that desired property is well represented in the model's training set (Figure 5).

### Update Nov 25, 2024
Have read the other reviews and authors' responses. See comment I added to their rebuttal below.

**Strengths:**

# S1. Paper includes detailed descriptions of the model and approach through algorithms and figures.
Figure 3 and 4, provide a detailed visual description of the MolMiner architecture and I found these helpful to understand what was happening. (A couple of very minor comments: coefficient is misspelled in Figure 3, and the box showing Figure 3 inside Figure 4 is blurry at the resolution used).

# S2. Calibration plots are clear
Plotting the calibration curves for property optimization seems a nice way to visualize the results, and with the inclusion of the KDE show how performance depends somewhat on how "in-distribution" a property score at that value is.

# S3. Including 3D information into the 2D generation seems an interesting new idea
Including the 3D conformation information into the generation process seems interesting, and could potentially lead to better structures. While it would have been nicer to have seen better evidence for the idea's importance (e.g., via a suitable ablation study; see W1 below), this aspect of the model seems to differ from previous fragment based approaches.

**Weaknesses:**

# W1. Experiments do not demonstrate the approach's advantages compared to existing methods or explain design choices made (hence why low "soundness" score).
The experiments do not consider any baselines, such as the alternative fragment-based approaches discussed (such as Jin et al., 2019, 2020) or non-fragment-based approaches (such as Gómez-Bombarelli et al., 2018). It is therefore hard to gauge how useful MolMiner would be and what advantages it offers over existing, already established approaches.
Moreover, I also have a hard time distinguishing the importance of the design choices made, e.g., the incorporation of 3D information  (lines 256-258) or the semi-order agnostic serialization method proposed (line 263). (Information on 3D information is shown in Table 2 of the appendix, but it seems to only be in terms of accuracy, rather than the effect on optimization, and the effect seems fairly minor).

# W2. The method for optimizing molecules (based on picking an initial fragment according to how well it is modeled to affect a particular property score) seems like it is unlikely to scale to more interesting optimization tasks.
My current understanding of the optimization method (lines 380-386) is that it involves learning a feed forward neural network to pick the initial fragment, but that the rest of the fragment generation process happens as usual. For harder optimization tasks, I worry that other parts of the molecule will also be important and so this method is unlikely to scale.

# W3. Novelty of method (modeling molecules via generating fragments) seems low and I would have liked to have seen the differences to these approaches better studied or discussed. (hence why low "contribution"/"presentation" score).
There are several existing approaches to generating molecules that take a fragment based approach. Examples include the cited Jin et al. (2019, 2020) as well as others such as:
- Marco Podda, Davide Bacciu, Alessio Micheli (2020) 'A Deep Generative Model for Fragment-Based Molecule Generation' AISTATS 2020.
- Bengio, E., Jain, M., Korablyov, M., Precup, D. and Bengio, Y. (2021) ‘Flow network based generative models for non-iterative diverse candidate generation’, NeurIPS 2021
- Xie, Y., Shi, C., Zhou, H., Yang, Y., Zhang, W., Yu, Y. and Li, L. (2021) ‘MARS: Markov Molecular Sampling for Multi-objective Drug Discovery’, ICLR 2021.
- Maziarz, K., Jackson-Flux, H., Cameron, P., Sirockin, F., Schneider, N., Stiefl, N., Segler, M. and Brockschmidt, M. (2022) ‘Learning to Extend Molecular Scaffolds with Structural Motifs’, ICLR 2022.
One of the main differences here seems to be the inclusion of the 3D information (see S3), but it would have been nice to have seen a comparison and discussion of these previous approaches.

**Questions:**

Other questions I had:
1. Lines 406-407 mentions accuracy (in the context of optimizing MolMiner’s hyperparameters). What does accuracy mean here? (Is this the fragment-level reconstruction accuracy?)
2. What is the "GetSSSR" function on line 110?
3. I'm a little confused as to the remapping procedure described in section 2.2. Why not just use atom map numbers in the SMILES to define the attachment points? (I think my main confusion is around how the symmetries are dealt with. Are all points that are symmetrically the same maintained as attachment points or is just one chosen?)
4. I just wanted to check if I understood the notation $\mathcal{U}(\mathcal{S}(\mathcal{M}))$ in Equation 1 correctly. Is this a uniform distribution over all possible stories? How important is this sampling for performance (compared to e.g. using a fixed ordering across all epochs, or restricting the number of possible orders by using a depth first search).
5. The datasets used seem quite small (~8500 molecules used for training). Does this limit the number of fragments observed and how big was the resultant fragment vocabulary?

---

> ### Author Response · Authors · 2024-11-19
> **Weaknesses 1, 2**
>
> We would like to thank the reviewer for its detailed comments. We hope we will clarify some points with our comments here, and we have addressed some of the raised issues in a revised version.
>
> # Weakness 1
>
> > Reviewer: "The experiments do not consider any baselines, such as the alternative fragment-based approaches discussed (such as Jin et al., 2019, 2020) or non-fragment-based approaches (such as Gómez-Bombarelli et al., 2018). It is therefore hard to gauge how useful MolMiner would be and what advantages it offers over existing, already established approaches."
>
> We have now benchmarked our model against HierVAE (Jin et al 2020), which shows that our model can generate significantly more novel molecules subject to desired conditions than its predecessor, HierVAE.
>
> > Reviewer: Moreover, I also have a hard time distinguishing the importance of the design choices made, e.g., the incorporation of 3D information (lines 256-258) or the semi-order agnostic serialization method proposed (line 263). (Information on 3D information is shown in Table 2 of the appendix, but it seems to only be in terms of accuracy, rather than the effect on optimization, and the effect seems fairly minor).
>
> Incorporating the 3D geometry is shown to contribute to better reconstruction accuracy during training, as shown in Table 2.Its effect is minor, but we suspect this is part of larger trend, for example, if one looks at the leaderboard of MatBench a benchmark for ML methods in predicting materials properties, one can see that for 6 out of 13 tasks models that only use composition (without 3d structure) lead. And even when they do not lead the task they perform surprisingly well. Therefore the effect of 3D structure there is also smaller than one might think. With this we mean this is a general phenomena in the field, and that is what the authors are  working currently on explaining (in a different work), but it falls outside the scope of this paper, in this paper we show that 3d geometry improves the accuracy of the model.
>
> The empirical justification of using a semi-order-agnostic choice can be seen in the new benchmarking against HierVae (Jin et al 2020). They don't use a semi-order-agnostic, so every time they train on the same sequence, and so the model does not learn to flexibly construct molecules which results in a significant lower novelty score (although this is a consequence as well of our way of working with symmetries of fragments).
>
> # Weakness 2:
>
> > Reviewer: The method for optimizing molecules (based on picking an initial fragment according to how well it is modeled to affect a particular property score) seems like it is unlikely to scale to more interesting optimization tasks. My current understanding of the optimization method (lines 380-386) is that it involves learning a feed forward neural network to pick the initial fragment, but that the rest of the fragment generation process happens as usual. For harder optimization tasks, I worry that other parts of the molecule will also be important and so this method is unlikely to scale.
>
> We disagree, this simple model is the best and simplest way to initialize the autoregressive process. This model’s task is to predict the probability of any of the fragments in the vocabulary of fragments appearing in a molecule, given a set of the molecules properties. The output of the model is then, for each of the fragments in the vocabulary, the probability of finding that fragment anywhere in a molecule with those properties. Then, if we take top-k fragments, that is, the fragments that the simple models is most confident they will appear anywhere in the molecule, then we can pass one of them to the autoregressive model, and because this models will decode the whole molecule from any starting point, in any order, then as long as the initial fragment is in the original molecule then the autoregressive model will be capable of reconstructing the molecule from it. In other words, the simple model does not need to guess all the fragments that will be in a molecule given some properties, as long as it gets 1 right, then the autoregressive model, given it can work with any order and any starting point, will take it from there. Its the simplest and more scalable way one can think of initializing an autoregressive model.
>
> Furthermore, the autoregressive process, what the reviewer refers to as ‘rest of fragment generation’ does not ‘happen as usual’, the full process is condition dependent, at every decision in this process the choice is subjected to the target conditions.We believe this is a misunderstanding. One can see how the conditions are embedded upon every step of the generation process from figures 3,4, (in purple) labeled as “Conditions”. Also it is discussed in lines 297-299 and 311-313 in the revised text (it was present also in the previous version).

---

> ### Author Response · Authors · 2024-11-19
> **Weakness 3**
>
> # Weakness 3
> > Reviewer: W3. Novelty of method (modeling molecules via generating fragments) seems low and I would have liked to have seen the differences to these approaches better studied or discussed. (hence why low "contribution"/"presentation" score).
>
> Its not the use of fragments that is novel with our work, but the way they are used, hence why how we treat the fragments to incorporate symmetries takes an important role on our paper (sections 2.1, 2.2 in the main text, sections A.1,A.2 in the appendix). To our knowledge this processing of fragments is novel,  and crucial because otherwise the vocabulary of fragments could contain duplicates.
>
> And thats exactly what happens in models like HierVae (Jin et al, 2020), from their original implementation https://github.com/wengong-jin/hgraph2graph/blob/master/data/logp04/vocab.txt#L30C1-L40C42   in their extracted vocabulary, lines 30-40 correspond to the vocabulary of attachments of the fragment ‘C1=CC=[NH+]C=C1’. Within all these different attachments of the fragment, there are duplicates. For example, a): 'C1=C[CH:1]=CC=[NH+]1' and b): 'C1=[NH+]C=C[CH:1]=C1' are the same attachment  configuration (note this are indicated by a number next to the atom e.g C:1), but its only because their model does not take into account the symmetries that this is possible. Now, if during training, the model predicts a.) but the ‘correct answer’ is b) that would prevent the model to learn correctly. So the vocabulary is unnecessarily larger (because it contains duplicates), and the presence of duplicates harms the learning process. Our model deals with this by incorporating the symmetries. This we believe is a novel contribution on its own and we also incorporate geometry during the generation which has remained a notable challengel. To address the related work the reviewer points out:
>
> - Marco Podda, Davide Bacciu, Alessio Micheli (2020) 'A Deep Generative Model for Fragment-Based Molecule Generation' AISTATS 2020.
>
> This is a generative model that operates on SMILES strings. The objective of this work was to increase the chemical validity of the generated structures and increase their diversity. From their Introduction:
>
> > ‘Our experimental evaluation shows that our model is able to perform on par with state-of-the-art graph- based methods, despite using an inherently least expressive representation of the molecular graph. Moreover, we show that generated compounds display similar structural and chemical features to those in the training sample, even without the support of explicit supervision on such properties.’
>
> The authors mention the limited expressiveness of using SMILES strings as the modeling approach, although they are capable of leveraging this representation to increase the chemical validity of the generated molecules. But perhaps more important, from their conclusion:
>
> >‘... we show that our contributions can increase validity and uniqueness rates of LM-based models up to the state of the art, even though an inherently less expressive representation of the molecule is used. As regards future works, we aim at extending this model for task like molecular optimization. This will require the design of novel strategies to maintain high uniqueness rates, while preserving smoothness in latent space. In addition, we would like to adapt the fragment-based paradigm to graph-based molecular generators’
> So the authors want, in future work, to adapt the approach to a more expressive representation of molecules, based on graphs.
>
> Therefore one can argue that this further highlights the novelty of our work, MolMiner, because it achieves chemical validity while using a more expressive representation for the molecule.
>
> - Bengio, E., Jain, M., Korablyov, M., Precup, D. and Bengio, Y. (2021) ‘Flow network based generative models for non-iterative diverse candidate generation’, NeurIPS 2021
> - Xie, Y., Shi, C., Zhou, H., Yang, Y., Zhang, W., Yu, Y. and Li, L. (2021) ‘MARS: Markov Molecular Sampling for Multi-objective Drug Discovery’, ICLR 2021.
> - Maziarz, K., Jackson-Flux, H., Cameron, P., Sirockin, F., Schneider, N., Stiefl, N., Segler, M. and Brockschmidt, M. (2022) ‘Learning to Extend Molecular Scaffolds with Structural Motifs’, ICLR 2022.
>
>  While these works help contextualize our model within current research in the field, they do not hinder the novelty of our approach.
> None of this methods include geometry in the generation process. We want to thank the reviewer for pointing this important work to us. We have now added them to our revised version to contextualize our contributions.

---

> > ### Author Response · Authors · 2024-11-19
> > **Questions raised by the reviewer**
> >
> > # Questions
> >
> > > Lines 406-407 mentions accuracy (in the context of optimizing MolMiner’s hyperparameters). What does accuracy mean here? (Is this the fragment-level reconstruction accuracy?)
> >
> > Yes, That is correct, since the choice of the next fragment to dock is formulated as a classification problem, we report here the accuracy, the percentage of correct classifications out of the total.
> > _Note: In order to clarify this, we explicitly state now in lines 409-410  that the accuracy is the fragment-level reconstruction accuracy._
> >
> > > What is the "GetSSSR" function on line 110?
> >
> > We mention it in line 132, ‘Smallest Set of Small Rings (SSSR)’. So the function GetSSSR refers to the process of getting the Smallest Set of Small Rings. I want to note that this is not our notation, but the standard nomenclature for this process.
> > _Note: In order to further clarify and contextualize the use of SSSR we included a comparison with other fragmentation methods in  the appendix, section: “Comparison on fragmentation approaches”_
> >
> > > I'm a little confused as to the remapping procedure described in section 2.2. Why not just use atom map numbers in the SMILES to define the attachment points? (I think my main confusion is around how the symmetries are dealt with. Are all points that are symmetrically the same maintained as attachment points or is just one chosen?)
> >
> > Yes, one could use atom map numbers to keep track of the attachment points, for example, that is the case in HierVAE, however, this would not solve the symmetry problems. Taking that same HierVAE as an example, from their original implementation https://github.com/wengong-jin/hgraph2graph/blob/master/data/logp04/vocab.txt#L30C1-L40C42   in their extracted vocabulary, lines 30-40 correspond to the vocabulary of attachments of the fragment ‘C1=CC=[NH+]C=C1’, if I now inspect all these different attachments of the fragment one finds duplicates. In this example, a): 'C1=C[CH:1]=CC=[NH+]1' and b): 'C1=[NH+]C=C[CH:1]=C1' are the same attachment  configuration (note this are indicated by a number next to the atom e.g C:1), but its only because their model does not take into account the symmetries that this is possible. Now, if during training, the model predicts a.) but the ‘correct answer’ is b) that would prevent the model from learning correctly. So the vocabulary is unnecessarily large (because it contains duplicates), and the presence of duplicates harms the learning process. Our model deals with this by incorporating the symmetries.
> > _Note: This discussion has been now included in the appendix, section: “Comparison on fragmentation approaches”_
> >
> > > “Are all points that are symmetrically the same maintained as attachment points or is just one chosen?)”
> > Yes, because we can identify using our procedure that they are the same attachment point, we treat it as a single one. Every time one of the symmetric attachments appears in a molecule while we fragment it, we know its equivalent to any of the other symmetries, so that if the model while rebuilding the model chooses any of the symmetric-equivalent attachments, we recognize that as the correct choice.
> >
> >
> > > I just wanted to check if I understood the notation U(S(M)) in Equation 1 correctly. Is this a uniform distribution over all possible stories? How important is this sampling for performance (compared to e.g. using a fixed ordering across all epochs, or restricting the number of possible orders by using a depth first search).
> >
> > Yes, S(M) is the set of all possible stories of a molecule M, and because we sample every story with equal probability, therefore U(S(M)) where U denotes the uniform distribution.
> > If one were to fix the ordering for all epochs (the single same story every time) then the model converges with fewer number of samples (as shown in the new benchmarking done) and the training process is faster, because stories do not need to be sampled every time. However this comes at a cost. The model will only see a single way of constructing each molecule, we believe this is in part the reason why our model is capable of generating significantly more novel molecules than its predecessor.
> >
> > One could, as the reviewer points out, create orderings/stories using DFS. This is something that we mention in lines 241-244. We chose not to, and instead used a graph traversal procedure with a random exploration queue, because its the most general form of graph traversal possible for our application. We believe that Imposing any particular policy on the order at which the molecule should be grown would be unjustified, and it would reduce the number of possible stories. We impose a single constraint on the graph-traversal procedure, because it is the only one we could not get rid off: from lines 241-242 _“A story can start from any fragment and can be grown in any particular order, only bound by having to dock new fragments to existing ones”_.

---

> > > ### Author Response · Authors · 2024-11-19
> > > **Continuation of Questions**
> > >
> > > (Continuation of Questions)
> > >
> > > > The datasets used seem quite small (~8500 molecules used for training). Does this limit the number of fragments observed and how big was the resultant fragment vocabulary?
> > >
> > > While the training data used is around 8500 molecules, it must be noted that the vocabulary of fragments is created using all the data, as is the case in HierVAE, JT-VAE,among others. This is the case, because every fragment has some assigned embedding, but if the model sees a fragment that is not in its vocabulary it will simply not be able to handle it, it will return an index out of range error.
> > > The number of distinct fragments the model found in our dataset is 34, now show in figure 9 under subsection “COMPARISON ON FRAGMENTATION APPROACHES” in the appendix. However must be noted that our vocabulary of fragments is the joint (fragment, attachment_point), in this case, taking into account all the possible unique docking points of the 34 fragments above  (we remove symmetries) then the final vocabulary size is 263.

---

> > > > ### Comment · Reviewer_Pmwq · 2024-11-26
> > > > **Thanks to authors for their rebuttal**
> > > >
> > > > I am grateful to the authors for their detailed response to my and the other reviewers' reviews.
> > > >
> > > > Thanks to the authors for clearing up some of my confusion about how the method worked (e.g., how the optimization information was included in every step through an embedding and how symmetry in fragments was properly accounted for). I also appreciate the inclusion of a modified HeirVAE baseline, which I think is a step in the right direction for improving the evaluation of MolMiner. However, I still do not believe the current evaluation (on tasks, datasets, and baselines) is sufficient for demonstrating the MolMiner's advantages over existing approaches (this point also seems to have been brought up by Reviewers SNaR and YR4n). The evaluations that do exist I find hard to interpret; for instance, I understand that some of the existing approaches do not incorporate geometric information, but the ablation in the paper (Table 2) suggests that this addition is not actually that important for performance, and so it's hard for me to judge how consequential a contribution this is.

---

### Official Review · Reviewer_SNaR · 2024-11-07

**Soundness:** 2
**Presentation:** 3
**Contribution:** 2
**Rating:** 5
**Confidence:** 4

**Summary:**

The paper introduces a fragment-based generative model, MolMine, for generative molecules in an autoregressive fashion. To train MolMiner, the paper suggests using an order-agnostic way by random sampling valid molecular stories.  The experiment results demonstrate the proposed model could generate the target molecule conditioned on the given prompts.

**Strengths:**

1. The presentation of the paper is good. I appreciate the efforts of presenting the key steps of the proposed methods with intuitive explanations and figures，e.g. Fig1/Fig 2/Fig 4.

2. The Molminer seems to involve a novel Transformer-based structure, which differs from the Graph Neural networks used by previous fragment-based autoregressive models. The proposed structure could hold potential for scaling.

3. The experiments show an interesting setting and demonstrate the model could be effective in the studied scenarios.

**Weaknesses:**

Though I appreciate the motivation and the interesting evaluation, I believe the paper needs a major revision to get accepted by top conferences such as ICLR.

1. The key technical contribution is not clear. I suggest the authors carefully rephrase the scoop and technical contribution of the proposed method. Specifically, I agree that the fragmentization of large molecules could be the key to enabling flexible generation.  In contrast, the five points of the introduction are not convincing to me.  For example, the paper states that the Molminer differs from the diffusion-based approach, e.g. HierDiff [1], in the fact that it could enable flexible node numbers during generation; I would consider this more as a different feature instead of the benefits. For example, HierDiff[1] also states the motivation of non-autoregressive generation over autoregressive generation in the sense of global modeling, etc. Unfortunately, I do not see any empirical evidence to compare and justify any of the proposed advantages.


2. The evaluation parts are not comprehensive enough. Specifically, there is no baseline to compare in the experiment section which makes it very hard for me to assess the model from a systemic work perspective. From a method-oriented perspective, I am quite confused over the novelty of several important components of the proposed framework. What is the exact difference between the introduced Fragmentization procedure in Algorithm 1 and the corresponding fragmentization of previous work such as JT-VAE/HierDiff? And the training objective is a widely applied approximation for graph modeling, then I would like to consider the key novelty that originates from the network structure parts. The key components of the networks is the geometry parts which differ from previous graph transformers, however, after carefully checking the ablation in Table 1. I find that with no-geometry, the model could still perform very well, then what is the point of involving such information and increasing the complexity?

3. Some important related works for 2D fragment-based autoregressive generation of molecules are missing [2]

[1] Coarse-to-Fine: a Hierarchical Diffusion Model for Molecule Generation in 3D
[2] MARS: Markov Molecular Sampling for Multi-objective Drug Discovery

**Questions:**

Refer to Above

---

> ### Author Response · Authors · 2024-11-18
> **Weakness 1**
>
> Thanks a lot for the review. We value your input a lot. We believe there have been some missunderstandings, and so we will address here all the weaknesses and questions raised, together with a revised manuscript where we emphasize this discussion.
>
> # Weakness 1
> ## On the flexible node numbers during generation:
>
> _Note: The importance of enabling flexible node size during generation is now discussed in a new subsection in the appendix under the title  “On the importance of variable-size autoregressive generation of molecules”_
>
> Enabling flexible node numbers during generation signifies the removal of an arbitrary constraint. Why should any model predefine the size (in terms of atoms or in terms of fragments) of the molecule? We believe this is a result of adapting generative approaches from Vision tasks (image generation, where the image has a fixed size) to materials modeling without taking into account the unique challenges of materials.
>
> Given a set of target properties, the space of possible molecules that satisfy those properties do not necessarily have the same size, therefore, by constraining the space of possible solutions, to those with some predetermined size constrains the solution. Given that this is an unnecessary constraint (other than convenience), the justification should be demanded from models that impose it, and not of models that remove it.
>
> Finally and to further highlight the error that fixing the size of the molecule upon generation constitutes, we can go deeper into how the size of the molecule is actually chosen upon initialization in models that pre-define it. From [1]  section _C. Additional Experiments_ subsection _C.1. Experimental configuration._:
>
> > ‘In all non-autoregressive methods, the number of nodes used for sampling is drawn from the size distribution histogram calculated on the training set.’.
>
> So they sample the size of the molecule from the size distribution of the dataset. If they do so for non-conditional generation this imposes an unjustified constraint as discussed before. However if this is done for conditional generation (section _C.6. Conditional generation_) then this is altogether wrong. Because they are sampling from the distribution of sizes when they should be sampling from the conditional distribution of size given the property. In other words, **in general, given a set of target properties the distribution of sizes of molecules satisfying those properties IS NOT the general distribution of sizes**. We hope this highlights the fact that enabling the model to choose the size of the molecule during generation is the right approach.
>
> ## On the motivation of non-autoregressive generation over autoregressive generation in the sense of global modeling:
>
> We disagree on the claim of global modeling stated in [1] and pointed out by the reviewer. From [1]  they state
>
> >  ‘Compared with the autoregressive approach, non-autoregressive generative models are more promising for 3D molecule generation, due to their natural advantages of global modeling ability’.
>
>  It is unclear what they mean by global modeling. An autoregressive model can perfectly do global modeling, and in general it does. For example our model, MolMiner, does global modeling, in every decision the model takes as input the entire molecule grown up to that point, not just its local environment. But lets examine the citations [1] use to support the aforementioned statement [2][3][4][5]:
>
> From [2] in the introduction:
>
> > ‘The non-autoregressive approach successfully generates small molecular graphs in a very fast and efficient manner but suffers from difficulty in model training and a low validity ratio when generating larger graphs. Owing to such limitations, the autoregressive approach, which aims at sequentially generating a molecular graph node by node using an autoregressive distribution, has been a main research direction…. These methods successfully generate molecular graphs with a high validity ratio, while involves an iterative procedure for each generation which makes them less efficient.This work focuses on the non-autoregressive approach for fast and efficient generation of molecular graphs. Here we propose an efficient learning method to train a VAE that generates molecular graphs in a non-autoregressive manner.’
>
> From this extract we can not see supportive information for the claim of improved global modelling. The advantages of non-autorregressive models discussed in this extract come from the fact they are faster and more efficient according to the authors.
>
> (to be continued in the next comment)
>
> [1] Coarse-to-Fine: a Hierarchical Diffusion Model for Molecule Generation in 3D
>
> [2] Kwon, Y., Yoo, J., Choi, Y.-S., Son, W.-J., Lee, D., and Kang, S. Efficient learning of non-autoregressive graph
> variational autoencoders for molecular graph generation. Journal of Cheminformatics, 11(1):1–10, 2019.

---

> ### Author Response · Authors · 2024-11-19
> **(Continuation on Weakness 1)**
>
> (Continuation on Weakness 1)
>
> From [2] in their conclusion:
>
> > ‘While the non-autoregressive approach has the advantage of fast and computationally efficient generation of molecular graphs without any iterative procedure, the existing methods suffer from low performance. To overcome this limitation, we proposed an efficient learning method to build a graph VAE that generates molecular graphs in a non-autoregressive manner. In order to improve the generation performance, we introduced approximate graph matching, reinforcement learning, and auxiliary property prediction into the training of the model. ….. One downside of the proposed method its high complexity with regard to space and time. Both the space and time complexity increase with the size of the graphs, owing to the use of graph neural networks. Thus, the proposed method would be impractical for learning and generating large molecular graphs, e.g., hundreds of heavy atoms in a molecule.’
>
>  Once again the authors highlight the fast and efficient advantage, but no claim on global modelling advantage.
>
> From [3] : In the Conclusion
>
> > ‘Lastly, it will be promising to explore alternative generative architectures within the MolGAN framework, such as recurrent graph-based generative models (Johnson, 2017; Li et al., 2018b; You et al., 2018), as our current one-shot prediction of the adjacency tensor is most likely feasible only for graphs of small size.’
>
> Again there are no claims of better global modeling, and as a matter of fact, the authors seem to be pointing at autoregressive models (they refer to them as ‘recurrent’) as a promising direction to be able to deal with larger molecules.
>
> From [4] its a Normalizing Flow based generative model. There is no claim of improved global modeling over size-autoregressive models, instead the argument focuses on the equivariant nature of the approach.  Similarly,  [5] is a Diffusion based model. We could again not find any claims here about the advantage in global modeling of non-autoregressive models over autoregressive. Moreover, the only mentions of size-autoregressive models in [5] section Related Work:
>
> > ‘Tangentially related, other methods generate molecules in graph representation. Some examples are autoregressive methods such as (Liu et al., 2018; You et al., 2018; Liao et al., 2019), and one-shot approaches such as (Simonovsky & Komodakis, 2018; De Cao & Kipf, 2018; Bresson & Laurent, 2019; Kosiorek et al., 2020; Krawczuk et al., 2021). However such methods do not provide conformer information which is useful for many downstream tasks.’
>
> Point out that some of the autoregressive models do not provide conformer information, which is something that MolMiner does (incorporating 3D spatial information into the attention mechanism). And in the conclusion
>
> > ‘We presented EDM, an E(3) equivariant diffusion model for molecule generation in 3D. While previous non-autoregressive models mostly focused on very small molecules with up to 9 atoms, our model scales better and can generate valid conformations while explicitly modeling hydrogen atom’
>
> They compare the better scalability to larger molecules of their approach to other non-autoregressive approaches. This might well be because autoregressive approaches scale better to larger molecules, as also pointed out in the conclusion of [3], and so the comparison they do is only within non-autoregressive models. In any case, not any mention of the advantages in global modeling.
>
> In summary,
> On the flexible node numbers during generation: This constitutes an improvement over methods that do fix the size of the molecule upon generation, because constraining the size of the molecule is unjustified and the way the molecule size is constrained in conditional generation (e.g in [1]) is problematic.
>
> On the motivation of non-autoregressive generation over autoregressive generation in the sense of global modeling: The claims from [1], used to rebut our proposal, have been investigated and shown to be unfounded. We do not claim better global modeling over non-autoregressive approaches because we can't prove it, but neither does [1] prove the opposite.
>
> [1] Coarse-to-Fine: a Hierarchical Diffusion Model for Molecule Generation in 3D
>
> [2] Kwon, Y., Yoo, J., Choi, Y.-S., Son, W.-J., Lee, D., and Kang, S. Efficient learning of non-autoregressive graph
> variational autoencoders for molecular graph generation. Journal of Cheminformatics, 11(1):1–10, 2019.
>
> [3] De Cao, N. and Kipf, T. Molgan: An implicit generative model for small molecular graphs. arXiv preprint
> arXiv:1805.11973, 2018.
>
> [4] Satorras, V. G., Hoogeboom, E., Fuchs, F. B., Posner, I., and Welling, M. E(n) equivariant normalizing flows, 2022
>
> [5] Hoogeboom, E., Satorras, V. G., Vignac, C., and Welling, M. Equivariant diffusion for molecule generation in 3d, 2022.

---

> ### Author Response · Authors · 2024-11-19
> **Weakness 2**
>
> ## On the evaluation and baseline comparison in experiment section:
>
> _Note: New benchmarking done, it has been added to the revised version_
>
> In this new benchmarking done one can see how our model is capable of retaining comparable performance on conditional multi-property generation while significantly increasing the novelty ratio. We believe this result highlights the importance of our contributions.
>
> ## On the fragment extraction, difference with JT-VAE and HierDiff. :
>
> _Note: This discussion has been now included in the appendix section: “Comparison on fragmentation approaches”_
>
> Our approach of decomposing the fragments within a molecule is most similar with that of HierVAE, itself based on previous work by the same authors, JT-VAE. HierDiff is based on JT-VAE, so in summary our approach in MolMiner, HierVae, JT-VAE and HierDiff are closely related.
>
> Ours is the simplest among the others. It breaks a molecule into its smallest set of small rings (SSSR), and all bonds not within a ring and in the main text we provide the motivation to use such a simple approach. This also means that our fragments tend to be smaller than those of the rest of the approaches, but we found no issue with this (e.g the total number of fragments for the dataset used is 34). For example, two fused rings could constitute a single Fragment in HierVae, HierDiff or JT-VAE, whereas in MolMiner it would always be two fragments, one per ring. This is particularly useful because then, all fragments extracted by MolMiner are Single Cyclic graphs, which is what allows us to deal with Fragments symmetries in a way that neither HierVAE, HierDiff or JT-VAE do.
>
> As now shown in the new subsection of the appendix “COMPARISON ON FRAGMENTATION APPROACHES” if one does not take into account the symmetries, fragments contain duplicates, which harms the training process and unnecessarily increases the size of the vocabulary. We highlight this issue, which has to the best of our knowledge not been reported, on an example within the original implementation of HierVAE. We hope this further highlights the importance of the contributions and novelty of our work.
>
> ## On the novelty from the network structure parts.
> _Note this discussion on the role of 3D geometry is now added to the appendix subsection  “THE ROLE OF INCORPORATING GEOMETRY"_
>
> - **The training objective is not widely used**: Autoregressive models are not trained on different rollouts at every epoch, for every molecule. Moreover our rollouts are done in the most general way we could think of from lines 241-242 in the original text:
>
> > ‘A story can start from any fragment and can be grown in any particular order, only bound by having to dock new fragments to existing ones’.
>
>  This is the most general formulation the problem of autoregressive generation can take. And it's not a widely used formulation. We believe our contribution in this aspect is to generalize the autoregressive approach to its minimum constraint which allows us to formulate the problem as a semi-order-agnostic process and use the theory of order-agnostic processes.
>
> (to be continued in the next comment)

---

> ### Author Response · Authors · 2024-11-19
> **(continuation of Weakness 2)**
>
> (continuation to weakness 2)
>
> - __Our model does not operate on a graph__: While the chemical graph of a molecule is retained to perform chemical validity checks, the model architecture does NOT use the chemical graph. Its a transformer model, which can be viewed as a fully connected (dense) graph, so every fragment is connected to every other fragment (as opposed to what graph-based approaches for molecules often do, they follow only the chemical graph) and on top of that we embed the 3d geometry of the molecule in a way that makes it invariant to rotations and translations (the pairwise distance matrix). Therefore this is not a widely used graph based model.
>
> - __The embedding of the geometry and its role__: As the reviewer points out, this is a clearly novel part of our approach, but its not the only novel thing. We also note in appendix subsection “THE ROLE OF INCORPORATING GEOMETRY”  that the contribution of the 3D geometry, while positive, its not as great as one could have thought it would be. And we point out that future work will need to be done to explain this surprising result. However, and now we include it in that same section in the revised paper, this is not an Isolated phenomena. The unreasonable effectiveness of poor materials representations is a widely reported issue, including work by [Tian, S. I. P., Walsh, A., Ren, Z., Li, Q., & Buonassisi, T. (2022). What Information is Necessary and Sufficient to Predict Materials Properties using Machine Learning? https://arxiv.org/abs/2206.04968]. This widespread issue in Machine Learning for Materials and molecules is currently being studied, but in a global context of the field, which is out of the scope of this paper. In this paper we show that 3D geometry helps. Furthermore, if one wants to contextualize this one may take a look at matbench (https://matbench.materialsproject.org/) a benchmark of machine learning models for predicting materials properties. There, out of 13 tasks for general purpose algorithms, in 6 of them, the leading models do not use 3D geometry, and even in those where 3D geometry-aware models lead, composition restricted models seem to perform remarkably well, therefore and following the same argument the reviewer states, _why would we use 3D geometry at all?_. This we believe is the same thing we are reporting in our work. It requires further study, and we are actively working on it, but its global phenomena and so the scope its out of this work.

---

> ### Author Response · Authors · 2024-11-19
> **Weakness 3**
>
> # Weakness 3:
> We thank the reviewer for pointing this work out, we were unfamiliar with it. It proposes a different approach to multi-property conditional generation. We have now added it to the revised version to contextualize our contributions.

---

> > ### Comment · Reviewer_SNaR · 2024-11-25
> > **Response to Rebuttal**
> >
> > I appreciate the efforts of the authors in providing a response to my questions.
> > However, in the discussion over "On the flexible node numbers during generation", the authors mentioned a point that "In general, given a set of target properties the distribution of sizes of molecules satisfying those properties IS NOT the general distribution of sizes". The claim is true and actually in non-autoregressive approaches for molecule generation, such as EDM[1], the conditional generation experiment is exactly conducted by sampling size from the conditional distribution. Hence, this claim proposed could not serve as evidence for the limitations of the non-autoregressive approaches.
> >
> > Besides, after checking the "new benchmark", I believe it is still in a very initial state with limited metrics and benchmarks.
> >
> > I would increase my score for the efforts during rebuttal. However, given the major revision needed, I can not support the paper get accepted currently.

---

> > > ### Author Response · Authors · 2024-11-27
> > > **Response to Reviewer comment**
> > >
> > > We would like to thank the reviewer again for taking the time to discuss our work. We would like to address one more thing, following up on the discussion "On the flexible node number during generation", because its an interesting discussion nevertheless.
> > >
> > > HierDiff as far as we have seen, does not state they sample from the conditional distribution of sizes, which is what lead us to believe they did not, as the only mention of this procedure in more detail is done where they do unconditional generation, and there they sample the general distribution of molecular sizes. But perhaps its sensible to assume they did it as in EDM[1] as the reviewer points out, in order to check if that was the case, I tried looking at the source code of HierDiff, and found that the conditional script experiments have been lost (see: https://github.com/qiangbo1222/HierDiff/issues/5#issuecomment-1968788477) which didn't let me clarify this point.
> > >
> > >  But in case they did like EDM[1], and we missed it, I would like to mention some shortcomings of such approach. From EDM[1], section E
> > >
> > > > "We compute c, M ∼ p(c, M ) on the training partition as a parametrized two-dimensional categorical distribution where we discretize the continuous variable c into small uniformly distributed intervals."
> > >
> > > Then it becomes clear that this approach is really sensible to the choice of the bins-size used to discretize the distribution, and it will scale poorly when doing simultaneously several properties, where one would need to choose a sensible choice of the bin size for every condition used, which will be generally different.
> > >
> > > Thats why we believe the choice of the molecular size should be jointly determine by the model as it creates the molecule. In a general sense, separating the size of the molecule from the generation process is an artificial and unjustified division. This has been more recently the reason for new proposals of trans-dimensional diffusion processes, from [a]: _"Since the diffusion occurs in a fixed dimensional space, there is no way for the guidance to appropriately guide the dimension of the generated datapoint. This can lead to incorrect generations for datasets where the dimension greatly affects the nature of the datapoint created, e.g. small molecules have completely different properties to large molecules"._
> > >
> > > [a] Campbell, A., Harvey, W., Weilbach, C., Bortoli, V. D., Rainforth, T., & Doucet, A. (2023). Trans-Dimensional Generative Modeling via Jump Diffusion Models. https://arxiv.org/abs/2305.16261

---

### Author Response · Authors · 2024-11-19
**Revised version. Description of changes**

We have now submitted a revised version, where we have addressed the reviews received. The core changes can be divided in: 1.) A comparison between our proposed model and its predecessor, for better evaluation. 2.) We have added other related work, to further contextualize our contributions 3.) Emphasis or more clarity on some points raised by the reviewers that we believe can be misunderstandings

1.) The comparison of our model against its predecessor is presented and discussed in lines 442-474, with the new Table 1 and Figure 5. Further discussion is also found in the appendix, section _A.7 BENCHMARKING MOLMINER: CALIBRATION , NOVELTY AND SAMPLE EFFICIENCY_ , lines 1054-1097.

2.) The addition of other related work can be found in the Introduction.

3.) Enhanced clarity, further discussion:

- __Comparison to other fragmentation approaches__ can be now found in section _A.5 COMPARISON ON FRAGMENTATION APPROACHES_ (line 898) In this section we also visualize the fragments extracted by MolMiner (figure 9) and highlight what we believe is an unreported problem of the fragments extracted by other methods like HierVAE (Jin et al 2020) (figure 10)

- __The importance of enabling variable size during molecular generation__. This discussion can be now found in section _A.6
O N THE IMPORTANCE OF VARIABLE - SIZE AUTOREGRESSIVE GENERATION OF
MOLECULES_ (line 1026). Here we elaborate on the problematic nature of fixing the size of the molecule during generation.

Finally, we have modified the the conclusions to further emphasize the contributions of our work. line 478.

We hope that these changes address the issues raised by the reviewers and further highlight the importance and novelty of our work.

---

### Meta-Review · Area_Chair_cHgp · 2024-12-22

**Metareview:**

In this work, authors introduce MolMiner, a transformer-based autoregressive model for generating molecules by assembling fragments. The key technical contributions include: (1) A fragment-based molecular generation approach that ensures chemical validity, (2) Incorporation of 3D geometric information into the generation process, (3) A semi-order-agnostic training procedure that allows flexible generation orders.
The paper demonstrates reasonable performance on multi-property optimization tasks involving solubility, redox potential, and synthetic accessibility. The authors show calibration curves indicating the model can effectively bias generation toward desired property values.
The presentation is generally clear, with detailed algorithms and figures explaining the architecture and generation process. The authors put significant effort into explaining technical details of fragment processing and symmetry handling.

However, the work has several critical weaknesses as follows. First, as noted by Reviewers SNaR and Pmwq, the experimental evaluation lacks comprehensive comparisons to existing methods. While the authors added a comparison to HierVAE in rebuttal, this single baseline on a limited set of metrics is insufficient to demonstrate clear advantages over the state-of-the-art. The evaluation would be strengthened by comparing against a broader set of baselines on standard benchmarks. Second, the novelty of the core technical contributions remains unclear. As Reviewer YR4n points out, fragment-based molecular generation has been extensively studied. While the authors argue their symmetry handling and 3D geometry incorporation are novel, the ablation studies (noted by Reviewer Pmwq) suggest the 3D geometry provides only minor benefits. The authors' rebuttal acknowledges this limitation but argues it reflects broader challenges in the field. Third, the optimization approach appears potentially limited in scalability. As Reviewer Pmwq notes, the method of selecting initial fragments based on desired properties may not extend well to more complex optimization tasks. While the authors argue their approach is principled, empirical evidence of its effectiveness on harder problems would strengthen this claim.

The positive aspects - including chemical validity guarantees, interpretable generation process, and reasonable performance on the evaluated tasks - are noteworthy. However, given the limited experimental validation and unclear advantages over existing methods, the current submission falls short of ICLR's bar for acceptance.

**Additional Comments On Reviewer Discussion:**

Please see the comments above.

---

### Decision · Program_Chairs · 2025-01-22

Reject